green chemistry/nanotechnology/biophysics

anti-larval activity, silver nanoparticles, green synthesis, *Azadirachta indica*, *Citrullus colocynthis*

**Author for correspondence:**
Muhammad Akram Raza
e-mail: akramraza.cssp@pu.edu.pk

This article has been edited by the Royal Society of Chemistry, including the commissioning, peer review process and editorial aspects up to the point of acceptance.

# Biosynthesis, characterization and anti-dengue vector activity of silver nanoparticles prepared from *Azadirachta indica* and *Citrullus colocynthis*

Shafqat Rasool[1], Muhammad Akram Raza[1],
Farkhanda Manzoor[2], Zakia Kanwal[2], Saira Riaz[1],
Muhammad Javaid Iqbal[1] and Shahzad Naseem[1]

[1]Centre of Excellence in Solid State Physics, University of the Punjab, Quid-e-Azam Campus, Lahore 54590, Pakistan
[2]Department of Zoology, Lahore College for Women University, Jail Road, Lahore 54000, Pakistan

(iD) MAR, 0000-0002-7895-5013

We report here biosynthesis of silver nanoparticles (AgNPs) using aqueous extracts of (i) *Azadirachta indica* leaves and (ii) *Citrullus colocynthis* fruit and their larvicidal activity against *Aedes aegypti*. The UV–Vis spectroscopy absorption peaks occurred in the range of 412–416 nm for *A. indica* AgNPs and 416–431 nm for *C. colocynthis* AgNPs indicating the silver nature of prepared colloidal samples. The scanning electron microscopy examination revealed the spherical morphology of both types of NPs with average size of $17 \pm 4$ nm (*A. indica* AgNPs) and $26 \pm 5$ nm (*C. colocynthis* AgNPs). The X-ray diffraction pattern confirmed the face-centred cubic (FCC) structure with crystallite size of $11 \pm 1$ nm (*A. indica* AgNPs) and $15 \pm 1$ nm (*C. colocynthis* AgNPs) while characteristic peaks appearing in Fourier transform infrared spectroscopy analysis indicated the attachment of different biomolecules on AgNPs. The larvicidal activity at different concentrations of synthesized AgNPs $(1–20 \, \text{mg l}^{-1})$ and extracts $(0.5–1.5\%)$ against *Aedes aegypti* was examined for 24 h. A concentration-dependent larvicidal potential of both types of AgNPs was observed. The $LC_{50}$ values were found to be 0.3 and $1.25 \, \text{mg l}^{-1}$ for *C. colocynthis* AgNPs and *A. indica* AgNPs, respectively. However, both extracts did not exhibit any notable larvicidal activity.

# 1. Introduction

Nanotechnology, owing to unique, extraordinary and incredible properties of materials at nanoscale, has revolutionized almost every field of science and technology. Especially, in arena of medical science and medicine, it is considered as next logical step and the future medicine [1,2]. However, the potential risks such as harmful side effects on human and environment have made nanotechnology a double-edged sword [3,4].

Natural products-based synthesis of nanoparticles can help to reduce the hazards of nanotechnology and thus can boost its applications. Biosynthesis, based on green chemistry principles, is proactive approach and has numerous advantages over conventional chemical and physical methods such as cost-effectiveness, environment friendly, simple and safe, no use of hazardous chemicals as reducing agents, less wastage of materials, minimum energy usage, safer disposal and recycling [5,6]. Studies are also reported on the preparation of various types of nanostructures by using natural products such as vitamins, biodegradable polymers, enzymes, polysaccharides and different microorganisms including algae, fungi, bacteria and viruses. Nevertheless, plants extract-mediated synthesis of nanomaterials is considered advantageous due to stability of nanostructures, cost efficacy, availably of plants, lower contamination risks, require little maintenance, and simple to scale up as plants are nature's 'chemical factories' [6,7].

Silver nanoparticles are famous for their anti-bacterial, anti-viral and anti-fungal activates [8]. AgNPs are also used in textile industry, food packing, cosmetics, paints and detergents owing to their anti-microbial activity [9]. Preparation of AgNPs can be carried out by using extract of different parts of plant such as roots, stem, leaves or fruit. In the synthesis of AgNPs by plants, polyphenols play the main role in degradation of different organics compounds. These plant extracts usually play the dual role in the synthesis process as reducing and capping agent. During green synthesis, the coatings of various biomolecules on surfaces of AgNPs not only improve their stability but also enhance their biocompatibility by reducing the toxicity risks [8,9].

In this study, we chose *Azadirachta indica* and *Citrullus colocynthis* for the synthesis of AgNPs, because both of these plants are famous for good medicinal properties and exhibit effective biological activities. *Azadirachta indica* plant belongs to family of Meliaceae and has enormous uses in medical field from ancient times [10]. It is familiar as anti-bacterial, anti-fungal and anti-microbial plant as it has quercetin, β-sitosterol and nimbidin in their leaves and azadirachtin in its seeds [11]. The *A. indica* plant extract has been reported to increase the anti-microbial activity of extract-mediated nanoparticles [10,11]. *Citrullus colocynthis* (also known as bitter apple) is a member of Cucurbitaceae family. The *C. colocynthis* fruit contains many biomolecules including alkaloids, glycosides, flavonoids and fatty acids which are famous for important biological activities, such as anti-microbial, anti-oxidants, cytotoxic, anti-diabetic, anti-lipidemic and insecticidal [12]. That is why it is considered one of the best medicinal plants and is used in different biomedical applications including anti-microbial, anti-cancer activities [12–14].

According to World Health Organization (WHO), mosquitoes can be listed as one of the deadliest organisms because millions of humans die due to different diseases spread by them. Mosquitoes are the potential vector for many diseases like dengue, malaria, west Nile, chikungunya, Zika and yellow fever. Dengue fever incidence has increased 30-fold worldwide in the last 30 years and is transmitted by *A. aegypti* mosquito vector [15]. Treatment of these diseases is a challenge. To control the rapidly increasing mosquito-borne risks, researchers are working on different strategies. Since mosquitoes breed in water and at larval stage, it is easy to target them there. However, inhibiting larvae in water by conventional ways using pesticides can increase the toxic risks for environment and humans. Thus, natural pesticides such as plant extracts can be simple and side effect-free promising methodologies. The use of green chemistry-based nanomaterials can further enhance the effectiveness and efficiency of such approaches [16–19].

In this work, *A. indica* and *C. colocynthis*-mediated AgNPs were synthesized, characterized and tested for their larvicidal activity against *A. aegypti* and it was found that these AgNPs can potentially be considered good anti-larvicidal agents.

# 2. Material and methods

## 2.1. Material and chemicals

Fresh leaves of *A. indica* were collected from the Botanical Garden of University of the Punjab, Lahore, Pakistan and *C. colocynthis* fruit was obtained from local market of Lahore, Punjab, Pakistan. Both products were identified by the experts of Department of Botany, University of the Punjab, Lahore.

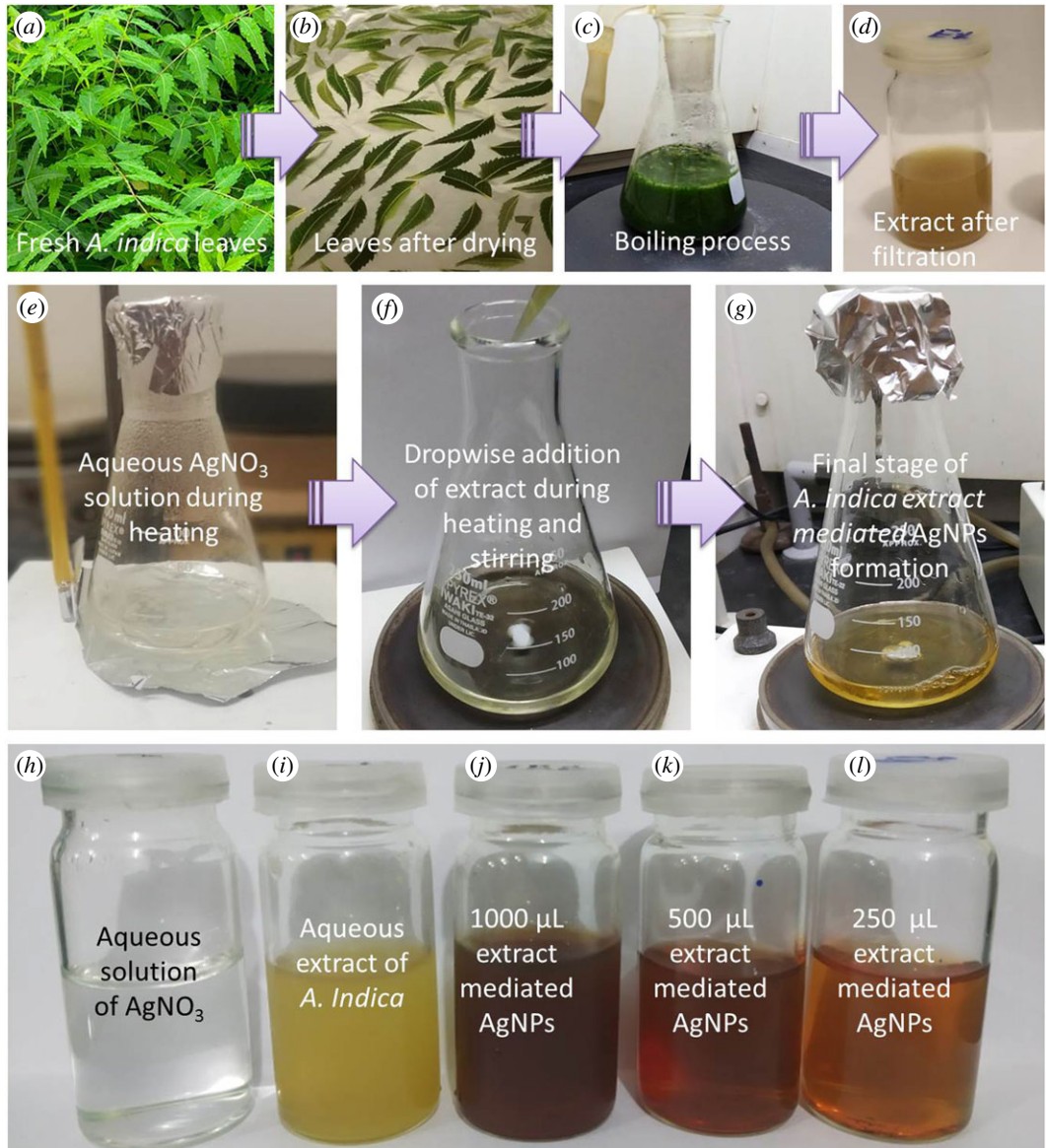

**Figure 1.** Upper panel: different steps during the preparation of aqueous extract of *A. indica* leaves; middle panel: synthesis of AgNPs; lower panel: (*h*) precursor solution, (*i*) reducing agent, (*j*–*l*) colloidal samples of AgNPs prepared by different amounts of *A. indica* extract.

Late fourth instar larvae of *A. aegypti* were collected from local pond and identified by the expert of the Department of Zoology, Lahore College for Women University, Lahore, Pakistan. Silver nitrate (AgNO$_3$) was of research grade and obtained from Merck (Germany). To prepare synthesis solutions, aqueous extracts and for all other purposes deionized (DI) water was used throughout the experiment.

## 2.2. Preparation of *Azadirachta indica* and *Citrullus colocynthis* extracts

The aqueous extract of *A. indica* leaves was prepared following the method reported by Pragyan *et al.* [20] with some modifications. To remove any dust and contaminations, first fresh leaves were rinsed carefully with running tap water then washed twice with DI water and were dried in air. Twenty grams of these leaves were cut into very small pieces which were transferred to 250 ml conical flask containing 100 ml DI water and boiled for 10 min in 100 ml DI water. After boiling, the mixture was cooled to room temperature and finally filtered with Whatman No. 1 filter paper before storing at 4°C for next step. Different steps of the preparation of *A. indica* leaves aqueous extract are shown in upper panel of figure 1. One can notice that greenish colour of mixture solution (figure 1*c*, before boiling) changed to yellowish light brown color in final stage (figure 1*d*).

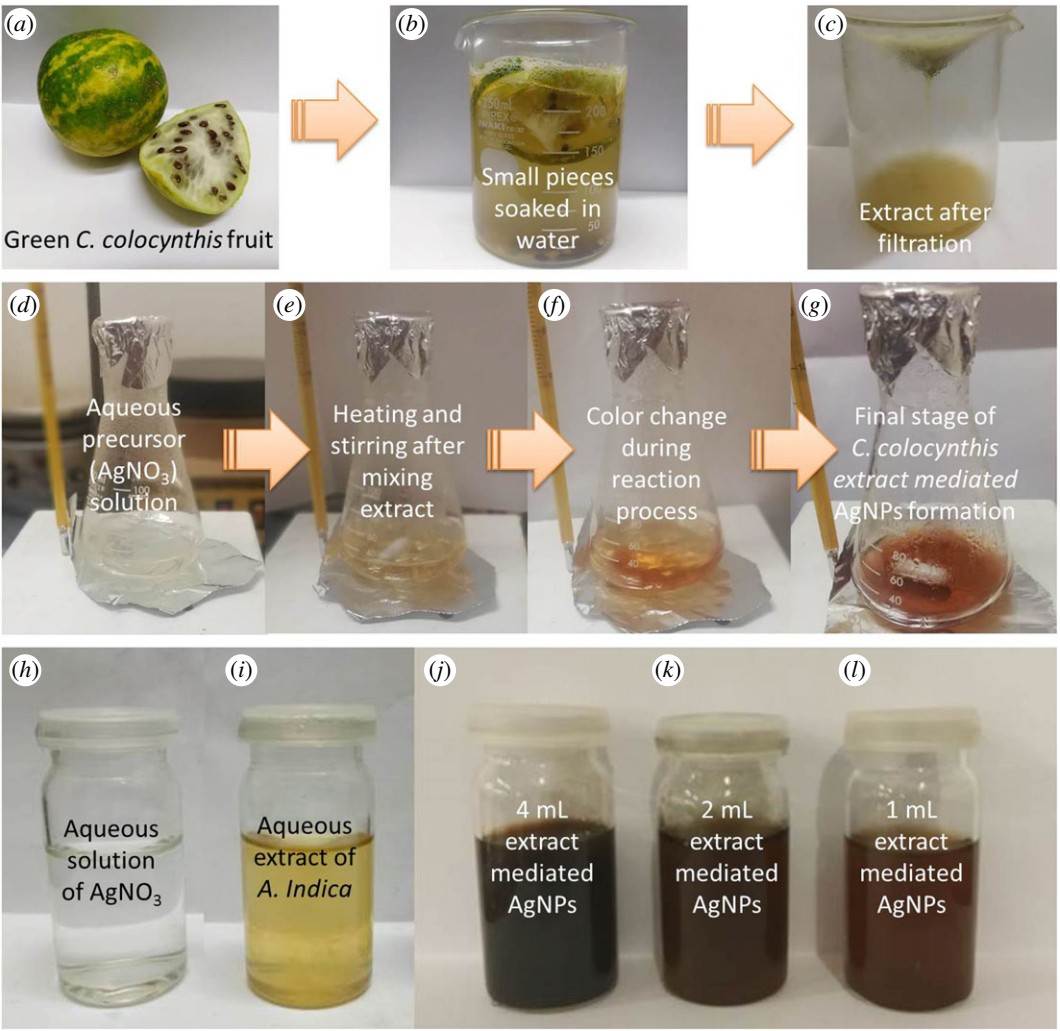

**Figure 2.** Upper panel: Different steps during the preparation of aqueous extract of *C. colocynthis* fruit; middle panel: synthesis of AgNPs; lower panel: (*h*) precursor solution, (*i*) reducing agent, (*j–l*) colloidal samples of AgNPs prepared by different amounts of *C. colocynthis* extract.

The aqueous extract of *C. colocynthis* fruit was prepared by the technique mentioned by Shawkey *et al.* [21] with few changes, and whole process of extract preparation is demonstrated in upper panel of figure 2. Briefly, a greenish white *C. colocynthis* fruit (figure 2*a*) was cleaned by washing many times with fresh tap water to get rid of any debris or any other contaminations, then rinsed twice with DI water before cutting into small pieces. Fifty gram pieces of *C. colocynthis* were soaked in 200 ml of DI water for 72 h at room temperature. Finally, mixture was filtered twice with Whatman No. 1 filter paper to get greenish yellow colour of the extract (figure 2*c*) and kept at 4°C for further use.

## 2.3. Synthesis of extract-mediated silver nanoparticles

To prepare AgNPs using aqueous extracts, first precursor stock solution (100 mM) was made by dissolving 1.69 g of silver nitrate (AgNO₃) into 100 ml of DI water.

*Azadirachta indica* extract-mediated AgNPs were made by the protocol of Aparajita & Mohan [22] with some amendments. Three different amounts of prepared extract (250, 500 and 1000 μl) were used for the same amount (20 ml) of silver nitrate precursor solution (1 mM) to obtain the optimized concentration. In figure 1 (middle panel), different phases of AgNPs preparation using the *A. indica* extract are presented. First of all colourless precursor solution of AgNO₃ was heated to 70°C with gentle magnetic stirring at 200 r.p.m. (figure 1*e*). Then *A. indica* extract was added dropwise into the solution and the mixture was kept for heating at hot plate at 70°C under continuous stirring for 10 min (figure 1*f*). The colour of the solution changed with time from transparent to light yellow to yellowish brown, indicating the formation

of AgNPs (figure 1g). At the end, the solution was cooled to room temperature and kept at 4°C for further use. Depending upon the amount of the extract (reducing agents) the colour of final stage sample (colloidal AgNPs) changed as can be noticed in lower panel of figure 1. However, only AgNPs prepared by using the 500 µl of extract were taken as optimized NPs based on UV–Vis results due to narrow peak width as compared with 250 and 1000 µl AgNPs and used in larvicidal activity and other characterizations.

For the synthesis of AgNPs using aqueous extract of *C. colocynthis*, approach described by Shawkey *et al.* [23] was followed with some alterations. Briefly, 20 ml of 5 mM silver nitrate solution was mixed with different amounts (1, 2 and 4 ml) of *C. colocynthis* extract and then the mixture was heated to the boiling temperature (about 95°C) under constant stirring at 200 r.p.m. for 20 min. Finally, solution was cooled down to room temperature and stored at 4°C. All three colloidal AgNPs samples mediated by different amounts of extract with precursor and extract samples are shown in figure 2 (lower panel). Nevertheless, only AgNPs medicated by 2 ml extract were selected as optimized NPs for larvicidal activity and other characterizations based on UV–Vis results (strong absorption intensity and narrow peak width as compared with 1 and 4 ml extracted-mediated AgNPs, respectively).

## 2.4. Characterization of synthesized silver nanoparticles

The above-prepared AgNPs were characterized by different techniques. Optical properties of all colloidal samples were examined by measuring the absorbance spectra with ultraviolet–visible (UV–Vis) spectroscopy (Shimadzu, UV-1800, Japan) in the wavelength range of 300–700 nm. The size, shape and elemental composition investigations of synthesized nanoparticles were conducted by scanning electron microscopy (SEM) and energy-dispersive X-ray (EDX; Nova NanoSEM 450, USA). For SEM analysis, samples were prepared by using the drop casting methods to achieve sufficient AgNPs amount on the clean glass substrate. To avoid any probable charging effects, a very thin gold coating was deposited on the samples before conducting SEM analysis. The structural nature of prepared NPs was examined by X-ray diffractometer (XRD; JSX 3201 M, Jeol, Japan). For XRD, again, samples were prepared by developing a thick enough film of each colloidal sample by drop casting methods on the glass substrate. To study the presence of different probable functional group on the surface of AgNPs, Fourier transform infrared (FTIR) spectroscopy (IRTracer-100, shimadzu, Japan) was conducted in the range of 500–3600 cm$^{-1}$.

## 2.5. Larvicidal activity tests

The larvicidal potential of both types of green AgNPs and their respective extracts at different concentrations against dengue larvae were studied according to the standard protocol recommended by WHO [24]. Twenty-five fourth instar *A. aegypti* larvae collected from local pond were transferred into beakers containing 200 ml DI water. Five different concentrations of both types of AgNPs were prepared and tested in each case; for *A. indica*-mediated AgNPs following concentrations were used: 1.25, 2.5, 5, 10 and 20 mg l$^{-1}$, and for *C. colocynthis*-mediated AgNPs concentrations used were: 0.3, 0.6, 1.25, 2.5 and 5 mg l$^{-1}$. The test concentrations of both types of AgNPs were selected on the basis of pilot experiments in which LC$_{50}$ concentrations (1.25 and 0.3 mg l$^{-1}$ for *A. indica* and *C. colocynthis*-mediated AgNPs, respectively) were found. In each beaker, the required amount of AgNPs was added to achieve the predetermined concentration. In the case of extracts, three concentrations (0.5%, 1% and 1.5%) of each type of extract were chosen, while DI water was used as control. The experiment was conducted in triplicate under laboratory conditions (at 26°C). All the beakers were covered with perforated aluminium foil for air circulation. During exposure experiment, larvae were not provided with any food. After 24 h of treatment period, the counting of alive and dead larvae was made to calculate per cent mortality. Larvae which showed no motility after being disturbed with a needle were counted as dead. The data are shown as mean value and standard error of mean (mean + s.e.); and Microsoft Excel program was used to make the graphs. The percentage mortality was determined by using formula

$$\% \text{ mortality} = \frac{\text{number of dead larvae}}{\text{total number of larvae exposed}} \times 100. \qquad (2.1)$$

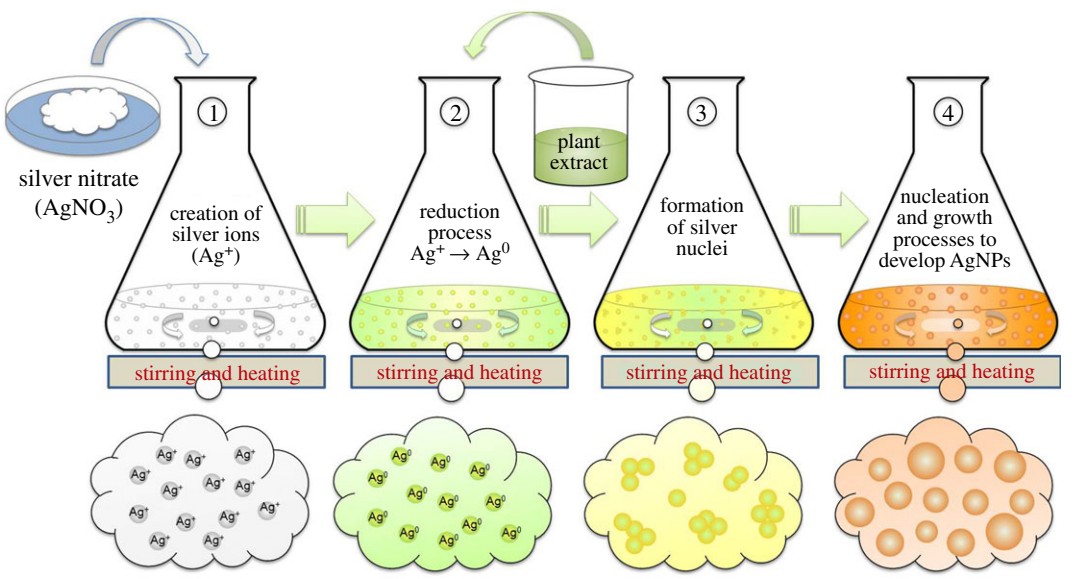

**Figure 3.** Schematic illustration of the synthesis mechanism of extract-mediated AgNPs. The phenomena occurring in each phase are mentioned accordingly.

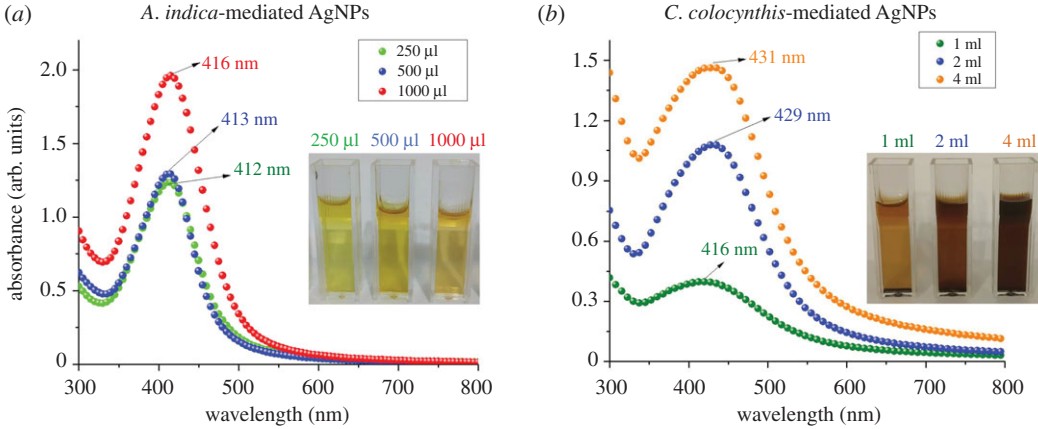

**Figure 4.** UV–Vis spectroscopy results of AgNPs synthesized by (*a*) three different volumes of *A. indica* leaves extract and (*b*) three different volumes of *C. colocynthis* fruit extract. Single absorption peak in each case appeared in the wavelength range of 412–430 nm indicating the spherical-shaped AgNPs. Insets display images of each colloidal sample in cuvette.

# 3. Results and discussion

## 3.1. Synthesis mechanism of extract-mediated AgNPs

Figure 3 illustrates schematically the potential formation mechanism of AgNPs using the aqueous extracts of different biomaterials. First of all, the precursor ($AgNO_3$) is dissolved into the solvent (DI water) that provides Ag-ions ($Ag^+$) into solution without producing any color. Addition of extract induces the reduction of these Ag-ions to free Ag-atoms ($Ag^0$) by gaining the electrons. This reduction can be termed bio-reduction or plant-assisted reduction, because it is caused by different biomolecules such as protein and phytochemicals present in the extract which serve as reducing agent [25]. This reduction reaction can be noticed by the colour change of the solution, which is enhanced by heating. The colour of the reaction solution may change from transparent to light yellow to yellowish brown and finally dark brown depending upon the type of the extract used (as shown in figures 3 and 4). This time-dependent colour change during the synthesis (10 min for *A. indica* AgNPs and 20 min for *C. colocynthis* AgNPs), actually, indicates the different phases of nucleation and growth process occurring in the reactions [26]. The free silver atoms ($Ag^0$) accumulate due to van der Waals interactions and Brownian motion to form Ag-nuclei (nucleation process). The growth process of these nuclei into AgNPs can

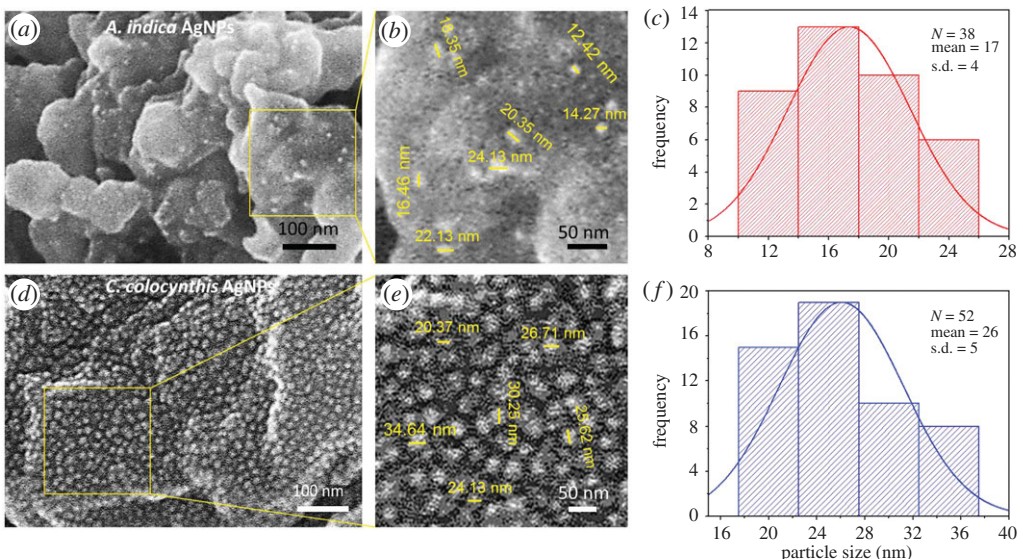

**Figure 5.** SEM micrographs and histograms. (*a*,*b*) SEM image of *A. indica* AgNPs showed spherical-shaped NPs decorated on larger extract particles. (*c*) Histogram of *A. indica* AgNPs indicated their diameter ranging between 12 and 24 nm. (*d*,*e*) SEM results of *C. colocynthis* AgNPs reveal the sphere-like morphology and random dispersal over the bigger extract particles. (*f*) Histogram of *C. colocynthis* AgNPs indicated their diameter ranging between 20 and 36 nm.

occur in different ways including coalescence of these nuclei and Ostwald ripening of smaller NPs into bigger ones. However, both nucleation and growth process may occur simultaneously in reaction during the formation of AgNPs [27]. The biomolecules of the extracts also served as capping agents because no additional stabilizing agent was used. The final size of the synthesized AgNPs can be controlled by varying the concentrations of extracts and precursor solution [28].

## 3.2. UV–Vis spectroscopy analysis

UV–Vis spectroscopy is a very common and effective way to study the formation of colloidal nanoparticles by their optical responses. In metallic NPs such as AgNPs, the free movement of electron between the valance band and conduction bands is possible due to very narrow gap between them. These electrons on the surface of AgNPs produce the surface plasmon resonance (SPR), which is actually resonant oscillation of conduction electrons in response of the incident light. Due to the SPR absorption, nanoparticles in colloidal form produce different colours depending upon various factors, e.g. size, shape and surrounding medium [25].

To study the optical response of colloidal AgNPs, UV–Vis spectroscopy was conducted, and obtained absorption spectra are displayed in figure 4. The characteristics SPR peaks of AgNPs using three different amounts of *A. indica* extract (250, 500 and 1000 µl) are shown in left panel (figure 4*a*). All three peaks were in the wavelength range of 412–416 nm indicating the pure silver nature of the particles. A slight red shift can be noticed with increasing concentration of the extract in the reaction solution. A similar trend of peak shifting towards the higher wavelength values (416, 429 and 431 nm) by enhancing the amount of extract (1, 2 and 4 ml) was also observed in the synthesis of *C. colocynthis*-mediated AgNPs as illustrated in right panel (figure 4*b*). This red shift might be due to the change in size of the AgNPs because, in green synthesis, extracts served as the reducing agents, and variation in reducing agent concentration can affect the final size of the nanoparticles [29,30]. Furthermore, the occurrence of only single absorption peak in each case indicates the formation of spherical-shaped AgNPs [31].

## 3.3. Scanning electron microscope analysis

SEM is the most prominent characterization technique to analyse shape, surface morphology, size distribution and elemental composition of nanomaterials. In order to examine the morphology of AgNPs prepared by both extracts, SEM analysis was conducted and the obtained micrographs with histograms of particle size distribution are presented in figure 5.

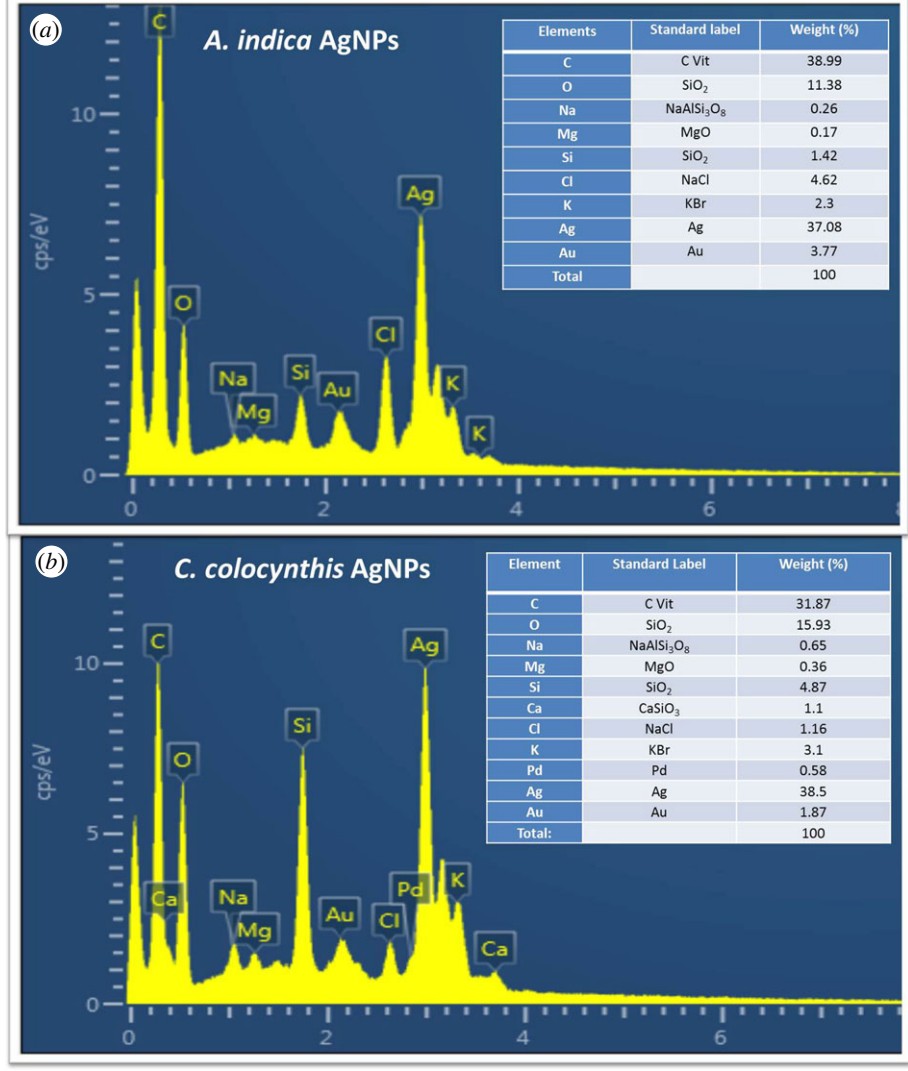

**Figure 6.** EDX spectrum showing elemental composition analysis of AgNPs. (*a*) *A. indica* leaves, (*b*) *C. colocynthis* fruit. Insets show the table of elements in each case.

In the case of AgNPs synthesized by *A. indica* extract, small spherical-shaped silver particles like bright tiny spots can be seen on the bigger bio-particles of the extract. At some places, agglomeration of few smaller particles into small clusters can also be noticed (figure 5*a*). The smaller nanoparticles have very high surface energy, and they agglomerate into larger-sized particles to minimize their surface energy. [32]. The size distribution range of these spherical AgNPs was found to be from 12 to 24 nm as depicted in a magnified view of figure 5*b* and presented graphically in particle size histogram (figure 5*c*). The average particle size was measured to be $17 \pm 4$ nm.

In the lower panel, SEM micrograph (figure 5*d*) revealed the size and morphology of AgNPs mediated from *C. colocynthis* extract. A nice random distribution of NPs over the bigger extract particles can be visualized. Again most of the NPs were found of spherical shapes with diameters in the range of 20–36 nm (figure 5*e*) and having average diameter of $26 \pm 5$ nm, as shown by the particle size histogram (figure 5*f*).

To confirm the existence of elemental silver in the synthesized particles, EDX was conducted. EDX technique determines the elemental composition analysis on the basis of energy values of characteristic X-rays which are unique for every element. The obtained EDX results are presented in figure 6, in upper panel for *A. indica* extract-mediated AgNPs while in lower panel for *C. colocynthis* extract-mediated particles. Insets show the table of elements in each case.

In both cases, we can notice strong silver signal (37.08% for *A. indica* AgNPs and 38.5% for *C. colocynthis* AgNPs) along with some other signals. The intense optical absorption peak at around 3 keV is considered a typical characteristic absorption band for metallic silver due to SPR [28]. Thus,

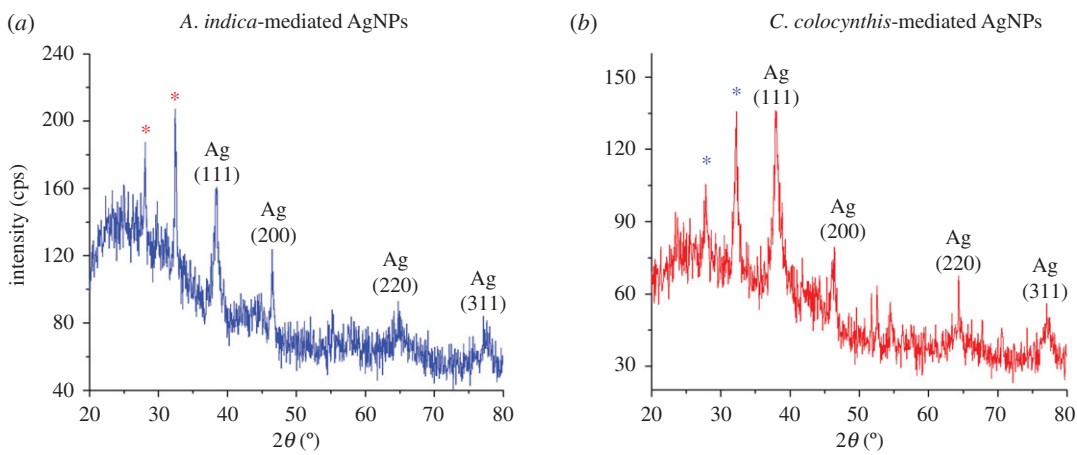

**Figure 7.** XRD patterns of (*a*) *A. indica* AgNPs and (*b*) *C. colocynthis* AgNPs.

EDX established the presence of elemental silver in both types of extract-mediated NPs by the characteristic SPR absorption peaks, which supported the UV–Vis results. The other signals such as C, O, Na, Mg, Au, Cl and K in the case of by *A. indica* AgNPs and C, O, Na, Ca, Mg, Si, Au, Cl, Pd and K (figure 6*a*) for the *C. colocynthis* AgNPs sample (figure 6*b*) can also be observed. The occurrence of weak gold (Au) signal can be attributed to thin gold coatings on the sample to avoid any charging. The Si peak may be due to the presence of glass substrate. The appearance of other signals, especially intense O and C peaks, indicates the presence of other metabolites on the AgNPs surface due to the aqueous extract (*A. indica* leaves and *C. colocynthis* fruit). These metabolites play a vital role of capping and stabilizing agents in the biosynthesis of the AgNPs, because they provide the stability to AgNPs by surrounding and developing a thin capping layer of organic molecules. This advantage of green synthesis approach not only reduces the cost but also minimizes the toxicity caused by hazardous chemicals used for reducing and capping puposes in chemical methods [28,33,34].

## 3.4. X-ray diffraction analysis

For the structural analysis of materials, X-ray diffraction (XRD) is an eminent analytical technique. It is widely used as primary analytical tool to investigate material purity and different structural parameters such as crystal structure, crystallite size, lattice parameter, crystal phase identification and various crystal defects [25]. In figure 7, indexed XRD patterns of both types of extract-mediated AgNPs are exhibited.

One can notice six prominent peaks in the diffractogram of *A. indica*-mediated AgNPs peaks (figure 7*a*). The peaks appearing at $2\theta$ values of 38.5°, 46.5°, 64.7° and 77.1° can be assigned to diffraction planes (hkl values) of (111), (200), (220) and (311), respectively, according to COD ID no. 9013052 [35]. These distinct reflection planes confirmed the silver metallic nature with face-centred cubic (FCC) crystalline structure of synthesized AgNPs. Moreover, two other unidentified intense peaks occurring at $2\theta$ values of 28.1° and 32.4°, showing higher degree of crystallinity, can also be noted (labelled by asterisks). These peaks indicate the presence of some crystalline moieties or organic compounds deposited on surface of AgNPs from aqueous extract of *A. indica* leaves. Many other studies of AgNPs prepared by green synthesis using the plant extract have also mentioned the appearance of such additional peaks in their XRD spectra which are believed to occur due to crystalline impurities of extracts [36,37].

In XRD pattern of AgNPs prepared by *C. colocynthis* as shown in figure 7*b*, four prominent distinct diffraction peaks observed at $2\theta$ value of 37.9° (111), 46.2° (200), 67.5° (220) and 77.1° (311) indicate the FCC crystal structure of resultant nanoparticle, according to COD ID no. 9013046 [35]. Again, some extra intense peaks at 27.8° and 32.2° were observed (asterisk-labelled peaks) which may be attributed to existence of crystalline nature of biomolecules on the AgNPs surface due to *C. colocynthis* extract [38].

To determine the crystallite size (*D*) of both type of extract-mediated AgNPs, the width of prominent Bragg's reflection (111) in each case was used in the Debye–Scherer formula [26].

$$D = \frac{k\lambda}{\beta_{hkl}\cos\theta} . \tag{3.1}$$

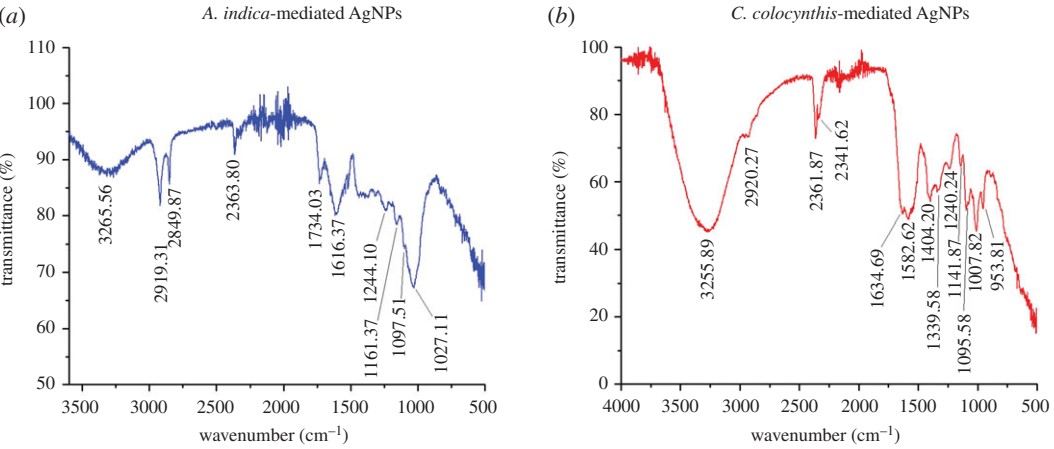

**Figure 8.** FTIR spectra of (a) A. indica AgNPs, (b) C. colocynthis AgNPs. Occurrence of various absorption peaks at different positions indicates the presence of different biomolecules on the surface of NPs.

**Table 1.** Calculated values of crystallite size and lattice parameter for both types of AgNPs.

| sample | peak position (2θ) | diffraction plane (hkl) | FWHM (rad) | crystallite size, D (nm) | lattice parameter, a (nm) |
|---|---|---|---|---|---|
| A. indica extract-mediated AgNPs | 38.5° | (111) | 0.0128 | 11 ± 1 | 0.403 |
| C. colocynthis extract-mediated AgNPs | 37.9° | (111) | 0.0095 | 15 ± 1 | 0.409 |

where $\lambda$, $k$, $\beta_{hkl}$ and $\theta$ are X-ray wavelength (1.54 Å), shape factor with value of 0.9, peak width (full width half maximum, FWHM) and Bragg diffraction angle, respectively. The average crystallite size determined by XRD for AgNPs prepared using extracts of A. indica and C. colocynthis were found to be $11 \pm 1$ nm and $15 \pm 1$ nm, respectively (table 1) which has fair agreement with particle size measurements obtained by SEM.

## 3.5. Fourier transform infrared spectroscopy analysis

FTIR spectroscopy is a very useful and powerful technique to identify different functional groups and chemical bonds in any sample. FTIR characterizes the chemical composition of the material by interaction of infrared radiation with a surface of the sample. The infrared radiations are absorbed by the sample at specific frequency ranges depending upon the type of chemical functional groups and bonds. The measurement of these frequencies and intensities helps to study the chemistry of the sample.

In this work, FTIR analysis was carried out, and obtained spectra are presented in figure 8. The aim of FTIR study was to examine the presence of different biomolecules and functional groups deposited on the surface of both types of AgNPs. These biomolecules helped in bio-reduction and efficient stabilization of NPs during the synthesis process. FTIR spectrum of AgNPs synthesized from A. indica leaves is shown in figure 8a, where various absorption bands at different specific frequency indicate a composite nature. The prominent peaks occurred at the frequency ranges of 3265, 2919, 2849, 2363, 1734, 1616, 1244, 1161, 1097 and 1027 cm$^{-1}$ as can be noticed from figure 8a. A broad band appearing in the range of 3265 cm$^{-1}$ can be attributed to the stretching vibrations of O–H bond indicating the presence of polyphenols. The peaks occurring at value range of 2919 and 2849 cm$^{-1}$ can be assigned to C=C aromatic and C–H alkaline (asymmetric stretching of C–H bonds) groups [39]. The possible presence of C≡N and C≡C triple bonds can be ascribed by peaks of 2363 cm$^{-1}$, because band appearing in the range of 2200–2400 cm$^{-1}$ indicates the C–N and C–C triples bonds. The band at 1734 cm$^{-1}$ could be due to C=O stretching vibration in the carbonyl groups from terpenoids and flavonoids. The peaks of 1616 cm$^{-1}$ might indicate presence of C=C bonds or aromatic rings attributed to carbonyl stretch in

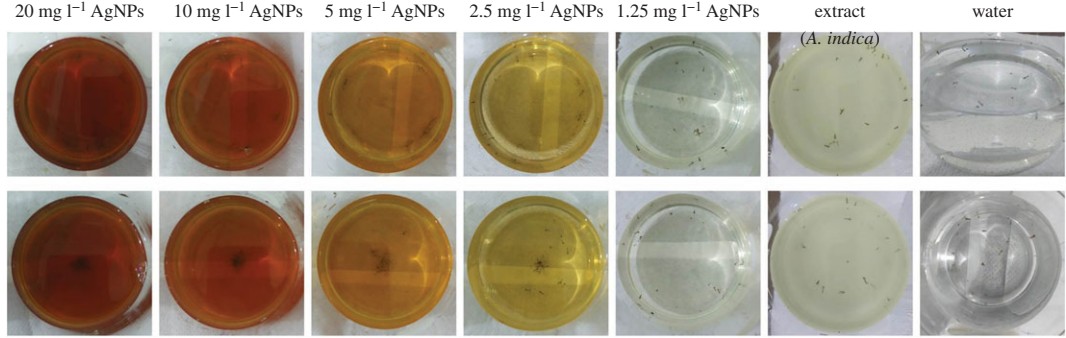

**Figure 9.** A representative display of larvicidal bioassay. Water (control), *A. indica* extract (conc: 1.5%), AgNPs at different concentrations. Upper panel shows the initial stage of experiment with 0 h time. Lower panel shows results after 24 h.

**Table 2.** Larvicidal activity of DI water (control), aqueous extract of *A. indica* leaves and extract-mediated AgNPs at different concentrations against fourth instars larvae of *A. aegypti*.

| sample type | concentration of sample | percentage mortality (%) |
|---|---|---|
| water (control) | — | 0 (active) |
| aqueous extract of *A. indica* leaves | 0.5% | 0 (active) |
| | 1% | 0 (active) |
| | 1.5% | 0 (less motile) |
| *A. indica* extract-mediated AgNPs | 1.25 mg $l^{-1}$ | 49 ± 4 |
| | 2.5 mg $l^{-1}$ | 66 ± 6 |
| | 5 mg $l^{-1}$ | 80 ± 4 |
| | 10 mg $l^{-1}$ | 85 ± 4 |
| | 20 mg $l^{-1}$ | 100 ± 0 |

proteins [40]. The absorption bands appearing at 1244, 1161, 1097 and 1027 $cm^{-1}$ can be assigned to the C–H alkene (aliphatic amines, C–H wag, C–N stretching in amide functional group) and C–O vibrations of ether linkages [14,39,41]. Thus, FTIR analysis of *A.indica*-mediated AgNPs confirmed the presence of different phytochemical components such as flavonoids, polyphenols and terpenoids of *A. indica* extract which potentially played a vital role in the reduction of Ag+ ions to $Ag^0$ atoms. Especially the proteins (due to the carbonyl and $NH_2$ of amino acid) present in *A. indica* leaves extract served as reducing and encapsulating agent because carbonyl group can reside on the metallic silver due to strong affinity for metals and thus developing a capping layer on surfaces of AgNPs [20,36,39,41].

For the AgNPs prepared by *C. colocynthis*, FTIR spectrum exhibited peaks at 3255, 2920, 2361, 2341, 1634, 1582, 1404, 1339, 1240, 1141, 1095, 1007 and 953 $cm^{-1}$ as shown in figure 8*b*. The occurrence of absorption bands at 953–1007 $cm^{-1}$ might be due to C–O– or C–O–C– functional groups while the peaks at 1010–1150 $cm^{-1}$ can be allocated to C–N stretching vibrations related to aliphatic amines or to alcohols or phenols indicating the existence of polyphenols [14,42]. The bands at 1240 and 1339–1460 $cm^{-1}$ are attributed to the amide III and II groups, respectively. Likewise the absorption in this region (around 1384 $cm^{-1}$) indicated the residual amount of $NO_3$. Whereas, the prominent absorption band at 1582 $cm^{-1}$ might be owing to symmetric stretching vibrations of –COO– groups of amino acid, confirming the presence of *C. colocynthis* extract protein. The existence of proteins was further indicated by the peak at 1634 $cm^{-1}$ due to stretch vibration of –C=C– of amide 1 bonds of proteins. The bands arising at around 2200–2400 $cm^{-1}$ indicate the probable presence of C–N or C–C triple bonds. The peaks occurring at 2920 $cm^{-1}$ can be dispensed to C–H bond vibrations. The intense broad absorption band at 3355 $cm^{-1}$ can be designated to characteristic of –OH (hydroxyl) group, indicating the existence of alcohols and phenolic compounds [34,43,44]. The FTIR results confirmed that the *C. colocynthis*-mediated AgNPs were surrounded by various phytochemical constituents such as amines, aldehydes, alcohols, ketones and carboxylic acids. These *C. colocynthis* extract-based biomolecules (proteins and metabolites) played dual role of bio-reduction and stabilization during the synthesis of AgNPs.

| 5 mg l⁻¹ AgNPs | 2.5 mg l⁻¹ AgNPs | 1.25 mg l⁻¹ AgNPs | 0.6 mg l⁻¹ AgNPs | 0.3 mg l⁻¹ AgNPs | extract (*C. colocynthis*) | water |

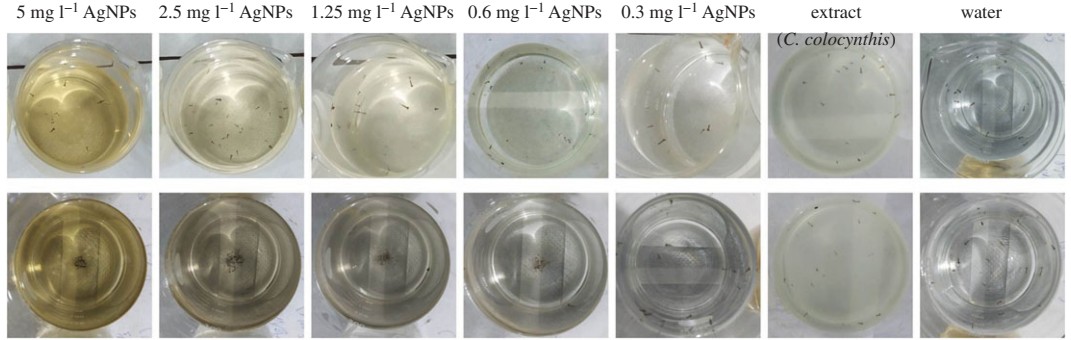

**Figure 10.** A typical demonstration of larvicidal activity. Water (control), *C. colocynthis* extract (conc: 1.5%), AgNPs at different concentrations. Upper panel shows stage in the start of the experiment (at 0 h). Lower panel displays the final stage story (after 24 h).

**Table 3.** Larvicidal results of DI water (control), aqueous extract of *C. colocynthis* fruit and extract-mediated AgNPs at different concentrations against fourth instars larvae of *A. aegypti*.

| sample type | concentration of sample | percentage mortality (%) |
| --- | --- | --- |
| water | — | 0 (active) |
| aqueous extract of *C. colocynthis* fruit | 0.5% | 0 (active) |
|  | 1% | 0 (less motile) |
|  | 1.5% | 0 (least motile) |
| *C. colocynthis* extract-mediated AgNPs | 0.3 mg l⁻¹ | 53 ± 4 |
|  | 0.6 mg l⁻¹ | 64 ± 6 |
|  | 1.25 mg l⁻¹ | 80 ± 3 |
|  | 2.5 mg l⁻¹ | 91 ± 3 |

## 3.6. Larvicidal bioassay

In order to study the anti-larval activity of aqueous extracts of *A. indica* and *C. colocynthis* and their respective AgNPs, different concentrations were tested for larvicidal bioassay against fourth instars larvae of *A. aegypti*. The typical demonstration of larvicidal bioassay are exhibited in figure 9 (*A. indica* panel) and figure 10 (*C. colocynthis* panel).

In the initial stage of this bioassay at $t = 0$ h, (as shown in upper panel of figure 9), random scattering of the larvae indicate their active movement in all the samples. After 24 h, lower panel of figure 9, all larvae in the control (water) were alive and moved energetically. Beakers containing extract of *A. indica* showed no mortality in all three concentration samples. Nonetheless larvae were found less motile in the sample with 3 ml concentration of extract as compared with other two samples (1 and 2 ml extract). Beakers containing AgNPs showed different mortalities with different concentrations. The dead larvae can be observed at the bottom or gathered in the middle of the each beaker with high concentration (lower panel of figure 9). AgNPs with concentration of 1.25, 2.5, 5, 10 and 20 mg l⁻¹ caused 49%, 66%, 80%, 85% and 100% mortality, respectively. All the larvicidal results of *A. indica* leaves extract and mediated AgNPs are summarized in table 2 and figure 11*a*.

A similar larvcidal activity assay was conducted for samples of *C. colocynthis* fruit extract; in this case, again three extract concentration; 0.5%, 1% and 1.5% and DI water as control were taken, while for extract-mediated AgNPs five concentration were chosen as 0.3, 0.6, 1.25, 2.5 and 5 mg l⁻¹ (figure 10). In the beginning of the activity test (at 0 h), all larvae were alive and motive in samples (upper panel of figure 10). After exposure of 24 h, again no mortality in control and all three extract samples was observed; however, a concentration-dependent effect on motility of larvae was noticed in extract-mediated NPs samples. The larvae in sample with 1.5% extract were least motile. In the case of AgNPs samples, highest mortality was counted in the sample with maximum concentration. A mortality of 53%, 64%, 80%, 91% and 100% were recorded for samples with AgNPs concentration of 0.3, 0.6, 1.25, 2.5 and

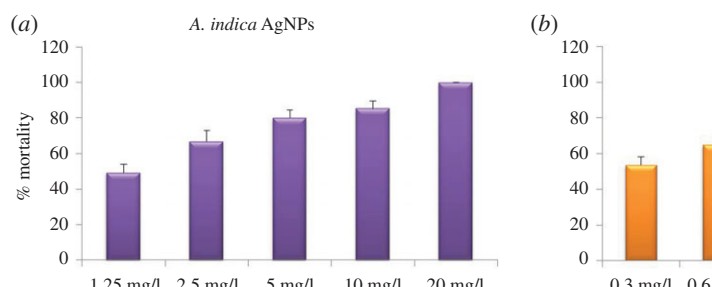
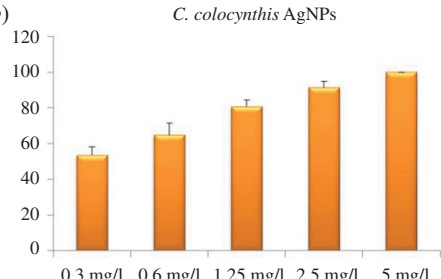

**Figure 11.** Graphs showing the larvicidal effect of AgNPs at different concentrations (*a*) *A. indica* AgNPs showed $LC_{50}$ at 1.25 mg $l^{-1}$ while at 20 mg $l^{-1}$ mortality was 100% (*b*) *C. colocynthis* AgNPs exhibited $LC_{50}$ at 0.3 mg $l^{-1}$ and 100% mortality was observed at 5 mg $l^{-1}$.

5 mg $l^{-1}$, respectively. Assembly of immotile (dead) larvae can be witnessed in middle of beakers (lower panel, figure 10). The results of *C. colocynthis*-mediated larvicidal assay are shown in table 3 and figure 11*b*.

We did not observe any larvicidal effect of both types of aqueous extracts (*A. indica* leaves and *C. colocynthis* fruit). Some other researchers have also reported negligible biocidal activity of aqueous extracts in comparison with extract-mediated nanoparticles [21,23,45]. On the other hands, a concentration-dependent biocidal efficiency of pure aqueous extracts of various natural products can also be found in the literature [16,28]. It seems that biocidal activities of extracts depend not only on the concentrations used in the bioassays but also on the level of purification (crude, highly purified, aqueous or alcoholic) as well as nature and different parts of biomaterials to be extracted such as leave, fruit, stem and seed [16,19,46].

In the case of extract-mediated AgNPs, the enhanced larvicidal activity of AgNPs can be attributed to the synergistic effect of Ag-ions and deposited biomolecules. The difference in the larvicidal efficiency occurs due to the presence of different types of biomolecules on the surfaces of AgNPs, though Ag-ions were same in both cases. We observed, in both cases, that by increasing the concentration of AgNPs, per cent mortality of larvae increased. As mentioned above, different types of biomaterials (plants) and even various parts of same plant may affect the biocidal efficiency of mediated AgNPs. A comparative larvicidal study of extracts and mediated AgNPs using various parts such as leaves, fruit, seed and stem of same plant can be conducted for more detailed investigations.

Since the exact mechanism of larvae mortality is unidentified, different modes of action can be proposed for larvicidal activity of AgNPs. Most probably the size and surface chemistry of AgNPs play a vital role to make them lethal against mosquito larvae. At the first stage, owing to small size, AgNPs penetrate into the larval membrane causing the possible leakage of cellular materials. In the next step, AgNPs may interact with the different cell molecules to inhibit moulting and other physiological processes. In their action, AgNPs may produce peroxide, inactivate the enzymes and disturb the functional proteins leading to the larvae death. The presence of extract biomolecules on the AgNPs surface can further enhance their effectiveness in this larvicidal process [23,39,45,46]. This indicates that anti-larval pharmaceuticals can be formulated using very low concentration of natural products-based AgNPs for practical applications. Thus, findings of this study can be helpful to provide new paradigm in designing the green chemistry-based alternative to combat the increasing mosquito-borne diseases.

## 4. Conclusion

AgNPs were prepared from herbal origin which is economical and eco-friendly technique using the aqueous extracts of *A. indica* leaves and *C. colocynthis* fruit. Both types of NPs were found pure metallic silver in nature having FCC crystalline structure and were spherical in shape with diameters of $25 \pm 5$ nm, as determined by SEM and XRD. The optical behaviour of colloidal samples and the presence of different extract biomolecules were studied by UV–Vis, and FTIR. Both types of prepared green AgNPs exhibited different larvicidal potential against *A. aegypti*; $LC_{50}$ at 1.25 mg $l^{-1}$ for *A. indica* AgNPs and $LC_{50}$ at 0.3 mg $l^{-1}$ for *C. colocynthis* AgNPs, respectively. We observed a concentration-dependent larvicidal activity in both types of AgNPs whereas their extracts showed negligible efficiency against larvae. These results suggest the potential use of green AgNPs as alternative to synthetic products in pharmaceuticals for anti-larval applications.

Data accessibility. Data available from the Dryad Digital Repository: https://doi.org/10.5061/dryad.z612jm68k [47].

Authors' contributions. S.R. and M.A.R. performed the synthesis experiments, analysed the data and wrote the manuscript. Z.K., S.R. and M.A.R. designed the experiments and conducted the larvicidal bioassays. S.R. and S.N. conducted the characterization measurements and helped in interpretation of data. F.M. and Z.K. provided the larvae facility and helped in larvicidal activity experiments. M.J.I. and S.R. conducted the FTIR and SEM characterization and helped in data analysis.

Competing interests. We have no competing interests.

Funding. This work was financially supported by Higher Education Commission (HEC) of Pakistan under the Project of National Research Program for Universities (Project no. HEC-NRPU-8019).

Acknowledgments. Authors acknowledge Dr Shahid Atiq and Dr Syed Sajjad Hussain from Centre of Excellence in Solid State Physics, University of the Punjab, Lahore, Pakistan for their cooperation in experimental work and useful discussions in manuscript writing.

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
