## [Reviewer comments · Royal Society Open Science]

Review History

RSOS-200540.R0 (Original submission)

Review form: Reviewer 1 (Mudassir Iqbal)

Is the manuscript scientifically sound in its present form?

Yes

Are the interpretations and conclusions justified by the results?

Yes

Is the language acceptable?

Yes

Do you have any ethical concerns with this paper?

No

Have you any concerns about statistical analyses in this paper?

No

Recommendation?

Accept with minor revision (please list in comments)

Comments to the Author(s)

The article describes the synthesis of Ag nanoparticle using *Azadirachta indica* and *Citrullus colocynthis* extracts. The work done is useful and is in line with the current scientific demands. There are also some issues to be addressed before publishing this work.

- There are several grammatical mistakes for instance I am highlighting some of these:
 - i) page 2 line 17 which have famous is incorrect it should be which are famous
 - ii) page 2 line 25 "vector for many deceases like" it should be vector for many diseases like
 - iii) page 2 line 25 please correct "fever.Dengue"
 - iv) page 2 line 35 "AgNPs were manufactured" were synthesized
 - v) page 2 line 35 "the mixture was let to cool down to room temperature" should be the mixture was cooled to room temperature

The authors should thoroughly check the whole manuscript and English should be improved as well as all the editorial mistakes should be rectified.

- Page 6 line 10, authors mentioned the particle size, I guess it should be crystallite size?
 - Authors have not mapped the particle size in SEM images. The particle size should be measured in SEM image and it should be visible in the picture.
 - The authors have mentioned that the chosen plants have known biological activity. However, they do not show any activity against *A. aegypti*. Secondly, if the extracts are not active against *A. aegypti* why there is a large difference in activity of AgNPs prepared from two different extracts as LC50 at 1.25 ppm for *A. indica* and LC50 at 0.3 ppm for *C. colocynthis* is observed. The later is 4 times more active than the former one. It does not seem that activity is only due to Ag NPs in that case it should be similar or close to each other.
 - Figures 1,2 and 5 are unnecessary and should be deleted.
 - the sentence in conclusion section "Both types of NPs were found pure metallic silver in nature having FCC crystalline structure and of spherical in shapes with diameters in the range of 25±5 nm, synthesized by different biomolecules of extracts as determined by different characterization techniques including SEM, UV- Vis, XRD and FTIR" is too long and ununderstandable. It should be refabricated to two sentences.
- After addressing these issues the article can be published in RSOS.

Review form: Reviewer 2

Is the manuscript scientifically sound in its present form?

No

Are the interpretations and conclusions justified by the results?

No

Is the language acceptable?

No

Do you have any ethical concerns with this paper?

No

Have you any concerns about statistical analyses in this paper?

Yes

Recommendation?

Major revision is needed (please make suggestions in comments)

Comments to the Author(s)

The MS presents work which has been published with various other plants. The work carried out is significant but need major revision.

MS needs enormous revision in scientific reporting, language and grammar. The scientific names are wrongly spelled at some places.

Each section of the MS needs major revision. The comments have been marked on the MS (Appendix A). Authors are advised to go through each and every comment; and rectify accordingly.

Decision letter (RSOS-200540.R0)

Dear Dr Raza:

Title: Biosynthesis, Characterization and Anti-dengue activity of Silver Nanoparticles prepared by *Azadirachta indica* and *Citrullus colocynthis*
Manuscript ID: RSOS-200540

The editor assigned to your manuscript has now received comments from reviewers. We would like you to revise your paper in accordance with the referee and Subject Editor suggestions which can be found below (not including confidential reports to the Editor). Please note this decision does not guarantee eventual acceptance.

Please submit your revised paper before 05-Jul-2020. Please note that the revision deadline will expire at 00.00am on this date. If we do not hear from you within this time then it will be assumed that the paper has been withdrawn. In exceptional circumstances, extensions may be possible if agreed with the Editorial Office in advance. We do not allow multiple rounds of revision so we urge you to make every effort to fully address all of the comments at this stage. If deemed necessary by the Editors, your manuscript will be sent back to one or more of the original reviewers for assessment. If the original reviewers are not available we may invite new reviewers.

RSC Associate Editor:
Comments to the Author:
(There are no comments.)

RSC Subject Editor:
Comments to the Author:
(There are no comments.)

Reviewers' Comments to Author:
Reviewer: 1

Comments to the Author(s)

The article describes the synthesis of Ag nanoparticle using *Azadirachta indica* and *Citrullus colocynthis* extracts. The work done is useful and is in line with the current scientific demands. There are also some issues to be addressed before publishing this work.

- There are several grammatical mistakes for instance I am highlighting some of these:
 - i) page 2 line 17 which have famous is incorrect it should be which are famous
 - ii) page 2 line 25 "vector for many deceases like" it should be vector for many diseases like
 - iii) page 2 line 25 please correct "fever.Dengue"
 - iv) page 2 line 35 "AgNPs were manufactured" were synthesized
 - v) page 2 line 35 "the mixture was let to cool down to room temperature" should be the mixture was cooled to room temperature

The authors should thoroughly check the whole manuscript and English should be improved as well as all the editorial mistakes should be rectified.

- Page 6 line 10, authors mentioned the particle size, I guess it should be crystallite size?
- Authors have not mapped the particle size in SEM images. The particle size should be measured in SEM image and it should be visible in the picture.
- The authors have mentioned that the chosen plants have known biological activity. However, they do not show any activity against *A. aegypti*. Secondly, if the extracts are not active against *A. aegypti* why there is a large difference in activity of AgNPs prepared from two different extracts as LC50 at 1.25 ppm for *A. indica* and LC50 at 0.3 ppm for *C. colocynthis* is observed. The latter is 4 times more active than the former one. It does not seem that activity is only due to Ag NPs in that case it should be similar or close to each other.
- Figures 1,2 and 5 are unnecessary and should be deleted.

- the sentence in conclusion section “Both types of NPs were found pure metallic silver in nature having FCC crystalline structure and of spherical in shapes with diameters in the range of 25 ± 5 nm, synthesized by different biomolecules of extracts as determined by different characterization techniques including SEM, UV- Vis, XRD and FTIR” is too long and ununderstandable. It should be refabricated to two sentences.

After addressing these issues the article can be published in RSOS.

Reviewer: 2

Comments to the Author(s)

The MS presents work which has been published with various other plants. The work carried out is significant but need major revision.

MS needs enormous revision in scientific reporting, language and grammar. The scientific names are wrongly spelled at some places.

Each section of the MS needs major revision. The comments have been marked on the MS. Authors are advised to go through each and every comment; and rectify accordingly.

Author's Response to Decision Letter for (RSOS-200540.R0)

See Appendix B.

RSOS-200540.R1 (Revision)

Review form: Reviewer 1 (Mudassir Iqbal)

Is the manuscript scientifically sound in its present form?

Yes

Are the interpretations and conclusions justified by the results?

Yes

Is the language acceptable?

Yes

Do you have any ethical concerns with this paper?

No

Have you any concerns about statistical analyses in this paper?

No

Recommendation?

Accept as is

Comments to the Author(s)

Nil

Review form: Reviewer 2

Is the manuscript scientifically sound in its present form?

Yes

Are the interpretations and conclusions justified by the results?

Yes

Is the language acceptable?

Yes

Do you have any ethical concerns with this paper?

No

Have you any concerns about statistical analyses in this paper?

No

Recommendation?

Accept with minor revision (please list in comments)

Comments to the Author(s)

I appreciate the efforts taken in revision of the manuscript. A few errors are yet to be addressed. The errors have been incorporated in the MS. Authors are requested to go through the MS (Appendix C) and address them.

Decision letter (RSOS-200540.R1)

Dear Dr Raza:

Title: Biosynthesis, Characterization and Anti-dengue vector activity of Silver Nanoparticles prepared by *Azadirachta indica* and *Citrullus colocynthis*
Manuscript ID: RSOS-200540.R1

Thank you for submitting the above manuscript to Royal Society Open Science. On behalf of the Editors and the Royal Society of Chemistry, I am pleased to inform you that your manuscript will be accepted for publication in Royal Society Open Science subject to minor revision in accordance with the referee suggestions. Please find the reviewers' comments at the end of this email.

The reviewers and handling editors have recommended publication, but also suggest some minor revisions to your manuscript. Therefore, I invite you to respond to the comments and revise your manuscript.

Because the schedule for publication is very tight, it is a condition of publication that you submit the revised version of your manuscript before 13-Aug-2020. Please note that the revision deadline will expire at 00.00am on this date. If you do not think you will be able to meet this date please let me know immediately.

Kind regards,
Dr Laura Smith
Publishing Editor, Journals

RSC Associate Editor:
Comments to the Author:
(There are no comments.)

RSC Subject Editor:
Comments to the Author:
(There are no comments.)

Reviewer comments to Author:
Reviewer: 1

Comments to the Author(s)
Nil

Reviewer: 2

Comments to the Author(s)
I appreciate the efforts taken in revision of the manuscript. A few errors are yet to be addressed. The errors have been incorporated in the MS. Authors are requested to go through the MS and address them.

Author's Response to Decision Letter for (RSOS-200540.R1)

See Appendix D.

Decision letter (RSOS-200540.R2)

Dear Dr Raza:

Title: Biosynthesis, Characterization and Anti-dengue vector activity of Silver Nanoparticles prepared from *Azadirachta indica* and *Citrullus colocynthis*
Manuscript ID: RSOS-200540.R2

It is a pleasure to accept your manuscript in its current form for publication in Royal Society Open Science. The chemistry content of Royal Society Open Science is published in collaboration with the Royal Society of Chemistry.

RSC Associate Editor
Comments to the Author:
(There are no comments.)

Reviewer(s)' Comments to Author:

Appendix A**ROYAL SOCIETY
OPEN SCIENCE****Biosynthesis, Characterization and Anti-dengue activity of
Silver Nanoparticles prepared by Azadirachta indica and
Citrullus colocynthis**

Journal:	Royal Society Open Science
Manuscript ID	RSOS-200540
Article Type:	Research
Date Submitted by the Author:	01-May-2020
Complete List of Authors:	Rasool, Shafqat; University of the Punjab, Quid-e-Azam Campus, Lahore-54590, Pakistan, Centre of Excellence in Solid State Physics Raza, Muhammad Akram; University of the Punjab, Centre of Excellence in Solid State Physics Manzoor, Farkhanda ; Lahore College for Women University, Department of Zoology Kanwal, Zakia; Lahore College for Women University, Jail Road, Lahore-54000,, Department of Zoology Riaz, Saira; University of the Punjab, Centre of Excellence in Solid State Physics Iqbal, Muhammad Javaid; University of the Punjab, Centre of Excellence in Solid State Physics Naseem, Shahzad ; University of the Punjab, Centre of Excellence in Solid State Physics
Subject:	Green chemistry < CHEMISTRY, Nanotechnology < CHEMISTRY, biophysics < CROSS-DISCIPLINARY SCIENCES
Keywords:	anti-dengue activity, silver nanoparticles, green synthesis, Citrullus colocynthis, Azadirachta indica
Subject Category:	Chemistry

Author-supplied statements

Relevant information will appear here if provided.

Ethics

Does your article include research that required ethical approval or permits?:

This article does not present research with ethical considerations

Statement (if applicable):

CUST_IF_YES_ETHICS :No data available.

Data

It is a condition of publication that data, code and materials supporting your paper are made publicly available. Does your paper present new data?:

Yes

Statement (if applicable):

Data Accessibility statement:

Our data are deposited at Dryad:

<https://datadryad.org/stash/dataset/doi:10.5061/dryad.z612jm68k> [48], with following reviewer URL;

Dryad Reviewer URL:

<https://datadryad.org/stash/share/DksW82FCDqISwPqI2aYhDI4C9QUE74hEBPTmrJ3trHw>

Conflict of interest

I/We declare we have no competing interests

Statement (if applicable):

CUST_STATE_CONFLICT :No data available.

Authors' contributions

This paper has multiple authors and our individual contributions were as below

Statement (if applicable):

Authors' Contributions:

- 1). SR and MAR performed the synthesis experiments, analysed the data and wrote the manuscript
- 2). ZK, SR and MAR designed the experiments and conducted the larvicidal bioassays
- 3). SR and SN conducted the characterization measurements and helped in interpretation of data
- 4). FM and ZK provided the larvae facility and helped in larvicidal activity experiments
- 5). SR and SN conducted the characterization measurements and helped in interpretation of data
- 6). MJR and SR conducted the FTIR and SEM characterization and helped in data analysis

Biosynthesis, Characterization and Anti-dengue activity of Silver Nanoparticles prepared by *Azadirachta indica* and *Citrullus colocynthis*

Shafqat Rasool¹, Muhammad Akram Raza^{*1}, Farkhanda Manzoor²,
Zakia kanwal², Saira Riaz¹, Muhammad Javaid Iqbal¹, Shahzad
Naseem¹

¹Centre of Excellence in Solid State Physics, University of the Punjab, Quid-e-Azam Campus,
Lahore 54590, Pakistan

²Department of Zoology, Lahore College for Women University, Jail Road, Lahore 54000,
Pakistan

Keywords: anti-dengue activity, silver nanoparticles, green synthesis, *Citrullus colocynthis*,
Azadirachta indica

Abstract

Green synthesis of nanomaterials using different bio-products can minimize the negative side effects of nanotechnology. Here, we report biosynthesis of silver nanoparticles (AgNPs) using aqueous extracts of (i) *azadirachta indica* leaves and (ii) *Citrullus colocynthis* fruit and their larvicidal activity against *Aedes Aegypti*. The prepared AgNPs were characterized by Fourier transforms infrared (FTIR), x-ray diffraction (XRD), uv-vis spectroscopy and scanning electron microscopy (SEM) for chemical, structural, optical and morphological analysis. Pure silver nature of colloidal NPs was indicated by occurrence of uv-vis absorption peaks in the range of 420-430 nm whereas XRD pattern confirmed the face centred cubic (FCC) structure of NPs. SEM examination revealed the spherical morphology of AgNPs with size 25±5 nm while characteristic peaks appearing in FTIR analysis indicated the attachment of different biomolecules on AgNPs. The anti-dengue activity of synthesized AgNPs at different concentrations (1ppm to 30 ppm) and as-prepared extracts against *Aedes aegypti* larvae were examined for 24 h. A concentration dependent larvicidal potential of both types of AgNPs was observed, however, *C. colocynthis* mediated AgNPs (LC₅₀ at 0.3 ppm) were found more effective at lower concentration than *A. indica* mediated AgNPs (LC₅₀ at 1.25 ppm). However, extracts did not exhibit any significant larvicidal activity.

1. Introduction

Nanotechnology, owing to unique, extraordinary and incredible properties of materials at nano-scale, has revolutionized almost every field of science and technology. Especially, in arena of medical science and medicine, it is considered as next logical step and the future medicine [1, 2]. However, the potential risks such as harmful side effects on human and environment has made nanotechnology a double-edged sword [3,4].

Natural products based synthesis of nanoparticles can help to reduce the hazards of nanotechnology and thus can boost its applications. Biosynthesis, based on green chemistry principles, is proactive approach and has numerous advantages over conventional chemical and physical methods such as cost-effectiveness, environment friendly, simple and safe, no use of hazardous chemicals as reducing agents, less wastage of materials, minimum energy usage, safer disposal and recycling [5, 6]. Studies are also reported on the preparation of various types of nanostructures by using natural products such as vitamins, biodegradable polymers, enzymes, polysaccharides and different microorganism including algae, fungi, bacteria and viruses. Nevertheless, plants extract mediated synthesis of nanomaterials is consider advantageous due to stability of

*Author for correspondence (Muhammad Akram Raza; Akramraza.cssp@pu.edu.pk).

†Present address: Centre of Excellence in Solid State Physics, University of the Punjab, Lahore-54590, Pakistan

nanostructures, cost efficient, availability of plants, lower contamination risks, require little maintenance, and simple to scale up as plants are nature's 'chemical factories' [6,7].

Silver Nanoparticles are famous for their anti-bacterial, anti-dengue and anti-fungal activities. AgNPs are also used in textile industry, food packing, cosmetics, paints and detergents owing to their anti-microbial activity. Preparation of AgNPs can be carried out by using extract of different parts of plant such as roots, stem, leaves or fruit. In the synthesis of AgNPs by plants, polyphenols play the main role in degradation of different organics compounds. These plant extracts usually play the dual role in the synthesis process as reducing and capping agent. During green synthesis, the coatings of various biomolecules on surfaces of AgNPs not only improve their stability but also enhance their biocompatibility by reducing the toxicity risks [8, 9].

In this study, we choose *Azedarachta Indica* (*A. Indica*) and *Citrullus Colocynthis* (*C. Colocynthis*) for the synthesis of AgNPs because both of these plants are famous for good medicinal properties and exhibit effective biological activities. *A. Indica* plant belongs to family of *Meliaceae* and has enormous uses in medical field from ancient times. It is familiar as anti-bacterial, anti-fungal and anti-microbial plant as it has quercetin, β -sitosterol and nimbinin in their leaves and azadirachtin in its seeds. The *A. Indica* plant extract has been reported to increase anti-microbial activity of extract mediated nanoparticles [10, 11]. *C. Colocynthis* (also known as bitter apple) is member of *Cucurbitaceae* family. It contains many biomolecules including alkaloids, glycosides, flavonoids, and fatty acids which have famous for important biological activities such as antimicrobial, antioxidants, cytotoxic, antidiabetic, antilipidemic, and insecticide. That's why it is considered one of the best medicinal plants and is used in different biomedical applications including anti-microbial, anticancer activities [12-14].

According to World Health Organization (WHO), mosquitoes can be listed as one of the deadliest organisms because millions of humans die due to different diseases caused by them. Mosquitoes are the potential vector for many diseases like dengue, malaria, west Nile, chikungunya, Zika and yellow fever. Dengue incidence has increased 30-fold worldwide in the last 30 years which is caused by dengue mosquito larvae; *Aedes aegypti* (*A. aegypti*) [15]. Treatment of these diseases is a challenge. To control the rapidly increasing mosquito-borne risks, researchers are working on different strategies. Since mosquitoes breed in water and at larval stage, it is easy to target them there. However, inhibiting larvae in water by conventional ways using pesticides can increase the toxic risks for environment and humans. Thus natural pesticides such as plant extracts can be simple and side effect free promising methodologies. The use of green chemistry based nanomaterials can further enhance the effectiveness and efficiency of such approaches [16-19].

In this work, *A. Indica* and *C. Colocynthis* mediated AgNPs were manufactured, characterized and tested for their anti-dengue activity and found that these AgNPs can potentially be considered good anti-larvicidal agents.

2. Materials and Methods

2.1. Material and chemicals

Fresh leaves of *A. indica* were collected from the Botanical garden of University of the Punjab, Lahore, Pakistan and *C. colocynthis* fruit was obtained from local market of Lahore, Punjab, Pakistan. Both products were identified by the experts of Department of Botany, University of the Punjab, Lahore. Late fourth instar larvae of *A. aegypti* were collected from local pond and identified by the expert of the Department of Zoology, Lahore College for Women University, Lahore, Pakistan. Silver nitrate (AgNO_3) was of research grade and obtained from Meck (Germany). To prepare synthesis solutions, aqueous extracts and for all other purposes deionized (DI) water was used throughout the experiment.

2.2. Preparation of *A. indica* and *C. colocynthis* extracts

The aqueous extract of *A. indica* leaves was prepared following the method reported by Pragyani et al [20] with some modifications. To remove any dust and contaminations, first fresh leaves were rinsed carefully with running tap water then washed twice with DI water and let them dry in air. 20g of these leaves were taken and converted into very small pieces by cutting then transferred to conical flask of 250ml to boil for 10 min in 100ml deionized water. After boiling, the mixture was let to cool down to room temperature and finally filtered with Whatman No. 1 filter paper before storing at 4 °C for next step. Different steps of the preparation of *A. indica* leaves aqueous extract are shown in Figure 1. One can notice that greenish color of mixture solution (figure 1c, before boiling) changed to yellowish light brown color in final stage (figure 1d).

The aqueous extract of *C. colocynthis* fruit was prepared by the technique mentioned elsewhere [21] with few changes and whole process of extract preparation is demonstrated in figure 2. Briefly, a greenish white *C. colocynthis* fruit (as shown in figure 2a) was cleaned by washing many times with fresh tap water to get rid of any debris or any other contaminations, then rinsed two times with DI water before cutting into small pieces. 50g pieces of *C. colocynthis* were soaked in 200ml of deionized water for 72 h. Finally mixture was filtered 2 times with Whatman No.1 filter paper to get greenish yellow color of the extract (figure 2c) and kept at 4°C for further use.

2.3. Synthesis of extract mediated silver nanoparticles

To prepare AgNPs using aqueous extracts, first precursor stock solution (100mM) was made by dissolving 1.69 g of silver nitrate (AgNO_3) into 100ml of DI water.

A. indica extract mediated AgNPs were made by the protocol of Aparajita et al. [22] with some amendments. We, in our case, used three different amounts of as-prepared extract (250 μl , 500 μl and 1000 μl) for the same amount 20 ml of silver nitrate precursor solution (1mM) to obtain the optimized concentration. In figure 3 (upper panel), different phases of AgNPs preparation using the *A. indica* extract are presented. First of all colorless precursor solution of AgNO_3 was heated to 70°C with gentle magnetic stirring of 200 rpm (figure 3a). Then *A. indica* extract was added dropwise into the solution and the mixture was kept for heating at hot plate at 70 °C under continuous stirring for 10 min (figure 3b). The color of the solution changed with time from transparent to light yellow to yellowish brown indicating the formation of AgNPs (figure 3c). At the end, the solution was let to cool down to room temperature and kept at -4°C for further use. Depending upon the amount of the extract the color of final stage sample (colloidal AgNPs) was found to change as can be noticed in lower panel of figure 3 (lower panel), from yellowish brown to reddish brown to reddish dark brown for 250 μl , 500 μl and 1000 μl extract mediated AgNPs respectively. However, only AgNPs prepared by using the 500 μl of extract was used in larvicidal activity and other characterization.

For the synthesis of AgNPs using aqueous extract of *C. colocynthis*, approach described by Shawkey et al. [23] was followed with some alterations. Briefly, First 20ml of 5mM silver nitrate solution was mixed with a defined amount of *C. colocynthis* extract then the mixture was heated to the boiling temperature (about 95°C) under constant stirring of 200 rpm for 20 min. Finally, solution was cooled down to room temperature and store at -4 °C. Color changes during the synthesis process indicate different phases of the synthesis and are presented in figure 4 (upper panel). We synthesized AgNPs using three amounts of as-prepared extract of *C. colocynthis* viz., 1ml, 2.5 ml and 4 ml following aforementioned method to achieve the optimized concentration. 
[revised manuscript text omitted]
 7a). There could be many reasons for agglomeration of these nanoparticles because these particles have very high surface energy and to reduce energy particles agglomerate and form larger sized particles [32]. The size distribution range of these spherical AgNPs was found to be from 12 nm to 24 nm (average particle size $\sim 17\pm 4$ nm).

In the lower panel, SEM micrograph (figure 9c) reveals the size and morphology of AgNPs mediated from *C. colocynthis* extract. A nice random distribution of NPs over the bigger extract particles can be visualized. No cluster formation due to agglomerate of NPs was noticed. Again most of the NPs were found of spherical shapes with average diameter of 26 ± 5 nm (figure 9d).

[revised manuscript text omitted]

Sample type	Concentration of sample	Percentage Mortality (%)
Water (control)	-	0 (active)
as -prepared	1 ml	0 (active)
aqueous extract of A. indica leaves	2 ml	0 (active)
	3 ml	0 (less motile)
A. indica extract	1.25 ppm	49±4
mediated AgNPs	2.5 ppm	66±6
	5 ppm	80±4
	10 ppm	85±4
	20 ppm	100±0

A similar larvicidal activity was conducted for samples of *C. colocynthis* fruit extract; in this case again three concentration of as-prepared aqueous extract; 1 ml, 2 ml and 3 ml and DI water as control were taken while for extracted mediated AgNPs five concentration were chosen as 0.3 ppm, 0.6 ppm, 1.25 ppm, 2.5 ppm and 5 ppm (figure 12). In the beginning of the activity test (at 0 h), all larvae were alive and motive in samples (upper panel of figure 12). After exposure of 24 h, again no mortality in (control) and all three extract sample, however, a concentration dependent effect on motility of larvae was noticed in extract samples. The larvae in sample with 3 ml extract were least motile. In the case of AgNPs samples, highest mortality was counted in the sample with maximum concentration. A mortality of 53%, 64%, 80%, 91% and 100% were recorded for samples with AgNPs concentration of 0.3 ppm, 0.6 ppm, 1.25 ppm, 2.5 ppm and 5 ppm respectively. Assembly of immotile (dead) larvae can be witnessed in middle of beakers (lower panel, figure 12). The statistics of *C. colocynthis* mediated larvicidal assay are listed in table 2.

Table 3 Larvicidal results of DI water (control), aqueous extract of *C. colocynthis* fruit and extract mediated AgNPs at different concentrations against fourth instars larvae of *A. aegypti*.

Sample Type	Concentration of sample	Percentage Mortality (%)
Water	-	0 (active)
as -prepared aqueous extract of C. colocynthis fruit	1 ml 2 ml 3 ml	0 (active) 0 (less motile) 0 (least motile)
C. colocynthis extract mediated AgNPs	0.3 ppm 0.6 ppm 1.25 ppm 2.5 ppm 5 ppm	53±4 64±6 80±3 91±3 100±0

In our case, larvicidal efficacy of both types of aqueous extracts (*A. indica* leaves and *C. colocynthis* fruit) was noticed very insignificant as compared to AgNPs mediated by these extracts. Some other researchers have also reported negligible biocidal activity of aqueous extracts in comparison to extracts mediated nanoparticles [21, 23, 47]. In other hands, a concentration-dependent biocidal efficiency of pure aqueous extracts of various natural products can also be found in literature [16, 28]. It seems that biocidal activities of extracts depend not only on the concentrations used in the bioassays but also on the level of purification (crude, highly purified, aqueous, or alcoholic) as well as nature and different parts of bio-materials to be extracted such as leave, fruit, stem and seed [16, 19, 46].

In both types of extract mediated AgNPs, We observed that by increasing the concentration of AgNPs, percent mortality of larvae increased. Nevertheless, *C. colocynthis* fruit extract mediated AgNPs were found more effective at lower concentrations and exhibited strong larvicidal efficacy as compared to *A. Indica* leaves extract mediated AgNPs as demonstrated graphically in figure 13. As mentioned above, different types of biomaterials (plants) and even various parts of same plant may affect the biocidal efficiency of mediated AgNPs. That's why perhaps AgNPs prepared by fruit extract of *C. colocynthis* showed better antilarval activity even at lower concentrations in our case. A comparative larvicidal study of extracts and mediated AgNPs using various parts such as leaves, fruit, seed and stem of same plant can be conducted for more detailed investigations.

Since the exact mechanism of larvae mortality is unidentified, different modes of action can be proposed for larvicidal activity of AgNPs . Most probably the size and surface chemistry of AgNPs play a vital role to make them lethal against mosquito larvae. At the first stage, owing to small size AgNPs penetrate into the larval membrane causing the possible leakage of cellular materials. In the next step, AgNPs may interact with the different cell molecules to inhibit molting and other physiological processes. In their action, AgNPs may produce peroxide, inactivate the enzymes and disturb the functional proteins leading to the larvae death. The presence of extract biomolecules on the AgNPs surface can further enhance their effectiveness in this larvicidal process [23, 40, 46, 47]. This indicates that anti-dengue pharmaceuticals can be formulated using very low concentration of natural products based AgNPs for practical applications. Thus findings of this study can be helpful to provide new paradigm in designing the green chemistry based alternative to combat the increasing mosquitos' diseases.

5. Conclusion

AgNPs were prepared from herbal origin which is economical and eco-friendly using the aqueous extracts of *A. indica* leaves and *C. colocynthis* fruit. Both types of NPs were found pure metallic silver in nature having FCC crystalline structure and of spherical in shapes with diameters in the range of 25±5 nm, synthesized by different biomolecules of extracts as determined by different characterization techniques including SEM, UV-Vis, XRD and FTIR. The prepared green AgNPs exhibited great larvicidal potential against *A. aegypti* even at low concentrations; LC₅₀ at 1.25 ppm for *A. indica* and LC₅₀ at 0.3 ppm *C. colocynthis* mediated AgNPs respectively. We observed a concentration dependent larvicidal activity in both types of AgNPs whereas their extracts showed negligible efficiency in both cases. These results indicate the potential use of green AgNPs as alternative to replace the synthetic products in pharmaceuticals for anti-dengue purposes.

Acknowledgments

Authors acknowledge Dr. Shahid Atiq and Dr. Syed Sajjad Hussain from Centre of Excellence in Solid State Physics, University of the Punjab, Lahore- Pakistan for their cooperation in experimental work and useful discussions in manuscript writing.

Ethical Statement

It is not relevant to our work.

Funding Statement

This work was financially supported by Higher Education Commission (HEC) of Pakistan under the Project of 'National Research Program for Universities' (Project No.: HEC-NRPU-8019).

Data Accessibility

Our data are deposited at Dryad: <https://datadryad.org/stash/dataset/doi:10.5061/dryad.z612jm68k> [48], with following reviewer URL;

Dryad Reviewer URL: <https://datadryad.org/stash/share/DksW82FCDqISwPqI2aYhDI4C9QUE74hEBPTmrJ3trHw>

Competing Interests

We have no competing interests.

Authors' Contributions

- 1). SR and MAR performed the synthesis experiments, analysed the data and wrote the manuscript
- 2). ZK, SR and MAR designed the experiments and conducted the larvicidal bioassays
- 3). SR and SN conducted the characterization measurements and helped in interpretation of data
- 4). FM and ZK provided the larvae facility and helped in larvicidal activity experiments
- 5). SR and SN conducted the characterization measurements and helped in interpretation of data
- 6). MJR and SR conducted the FTIR and SEM characterization and helped in data analysis

References

[revised manuscript text omitted]

Figure Captions

Figure 1. Different steps during the preparation of aqueous extract of *A. indica* leaves; (a) fresh leaves were collected, (b) leaves were kept under shade and dried on aluminum sheet (c) boiling of very small pieces *A. indica* leaves in DI water (d) *A. indica* extract in sample bottle after filtration

Figure 2. Different stages of *C. colocynthis* fruit extract preparation; (a) green fruit was cut into very small pieces, (b) small pieces soaked into DI water for 72 h at room temperature, (c) finally aqueous extracts were obtained by filtration.

Figure 3. (Upper panel) different phases of the synthesis of *A. indica* extract mediated AgNPs. (a) Formation of Ag-ions in the colorless precursor AgNO₃ solution during heating, (b) dropwise adding of extract during heating and constant stirring appearing of color indicate the reduction of Ag-ions, (c) Final stage color indicating the complete reduction of Ag-ions into AgNPs by the extract. (Lower panel) different samples in the bottles (a) transparent AgNO₃ solution, (b) yellowish green aqueous extract of *A. indica*, (c) colloidal sample of AgNPs mediated by 1000µl of extract, (d) AgNPs prepared by 500µl extract, and (e) AgNPs synthesized by 250µl of extract.

Figure 4. (upper panel) Different levels during the formation of AgNPs using the aqueous extract of *C. colocynthis* (a) transparent precursor solution, (b) solution after mixing the extract by stirring and heating, a slight color change to light yellowish brown, (c) further evident color change with time during heating and stirring indicate the reduction of Ag-ions to Ag-atoms and formation of AgNPs (d) completion of the reaction with reddish brown color of AgNPs. (lower panel) (a) AgNO₃ colorless precursor solution, (b) light yellow color *C. colocynthis* aqueous extract, (c-d) colloidal sample of AgNPs prepared by 4ml, 2.5 ml and 1 ml of *C. colocynthis* extract respectively.

Figure 5. Schematic illustration of synthesis mechanism of extract mediated AgNPs. The phenomena occurring in each phase in mentioned accordingly.

Figure 6. UV-Vis spectroscopy results of AgNPs synthesized by using; (a) three different concentrations of *A. indica* leaves extract, (b) three different concentration of *C. colocynthis* fruit aqueous extract. Single absorption peak in each case appearing in the wavelength range of 412 nm to 430 nm indicate the spherical shaped nano-sized silver particles. Insets display images of each colloidal sample in cuvette.

Figure 7. SEM micrographs and size distribution histograms of both types of biosynthesized AgNPs; (a) SEM image of *A. indica* mediated AgNPs showing the almost spherical shaped NPs decorated on the larger sized extract particles, (b) particle size histogram of *A. indica* AgNPs indicating the diameter range 12-24 nm. (c) SEM results of *C. colocynthis* mediated AgNPs revealing the sphere like morphology and nice random dispersal over the bigger extract particles. The average size of these AgNPs was found to be 26± 5 nm as can be seen in the histograms (d).

Figure 8. EDX spectrum showing elemental composition analysis of AgNPs prepared by the aqueous extracts of (a) *A. indica* leaves and (b) *C. colocynthis* fruit. Insets show the table of elements in each case. The presence of elemental silver and other biomaterials can be discerned by different absorption characteristics peaks in the both spectra.

Figure 9. X-ray diffraction patterns of (a) *A. indica* mediated AgNPs and (b) *C. colocynthis* mediated AgNPs. In both cases, along with characteristics peaks designating the face centered cubic (fcc) crystalline nature of prepared nanoparticles, appearance of few extra peaks (mentioned by stars *) indicate the presence of some extract molecules.

Figure 10. FTIR spectra of both types of AgNPs; (a) *A. indica* leaves extract mediated AgNPs, (b) AgNPs prepared using extract of *C. colocynthis* fruit. Occurrence of various absorption peaks at different positions in each case indicates the presence of difference biomolecules on the surface of synthesised nanoparticles.

Figure 11. A representative display of larvicidal bioassay for different test samples in the beakers including water control (water), *A. indica* extract and its mediated AgNPs at different concentrations (as mentioned); upper panel shows the initial stage of experiment with 0h time while results after 24 h are represented in lower panel. It can be noticed that in the beginning all larvae were active and randomly scattered but after 24h due to mortality most of the dead larvae are gathered in the middle of the beakers. However, in control, extract and samples with low AgNPs concentration scattered larvae can be discerned.

Figure 12. A typical demonstration of larvicidal activity of *C. colocynthis* extract mediated AgNPs with different concentrations, as prepared extract and water as control. In upper panel, different samples with larvae are shown in the start of experiment (at 0h) where random sprinkle of the larvae into the liquid can be seen. After 24h of exposure, larvicidal efficacy of different samples is presented in the lower panel where dead larvae can be found assembled at the bottom of the beakers.

Figure 13. Graphs showing the larvicidal performance in mortality (%) of both types of AgNPs at different test concentrations; (A) *A. indica* extract mediated AgNPs showed LC₅₀ at about 1.25 ppm while at 20 ppm 100% mortality was observed, (B) AgNPs prepared by *C.colocynthis* extract exhibited 50% mortality (LC₅₀) only at about 0.3 ppm and 100% mortality at 5ppm.

Figures

Figure 1

Figure 2

Figure 3

Figure 4

Figure 5

Figure 6

Figure 7

Figure 8

Figure 9

Figure 10

Figure 11

Figure 12

Figure 13

Figure 1. Different steps during the preparation of aqueous extract of *A. indica* leaves; (a) fresh leaves were collected, (b) leaves were kept under shade and dried on aluminum sheet (c) boiling of very small pieces *A. indica* leaves in DI water (d) *A. indica* extract in sample bottle after filtration.

227x71mm (150 x 150 DPI)

Figure 2. Different stages of *C. colocynthis* fruit extract preparation; (a) green fruit was cut into very small pieces, (b) small pieces soaked into DI water for 72 h at room temperature, (c) finally aqueous extracts were obtained by filtration.

194x77mm (150 x 150 DPI)

Figure 3. (Upper panel) different phases of the synthesis of *A. indica* extract mediated AgNPs. (a) Formation of Ag-ions in the colorless precursor AgNO₃ solution during heating, (b) dropwise adding of extract during heating and constant stirring appearing of color indicate the reduction of Ag-ions, (c) Final stage color indicating the complete reduction of Ag-ions into AgNPs by the extract. (Lower panel) different samples in the bottles (a) transparent AgNO₃ solution, (b) yellowish green aqueous extract of *A. indica*, (c) colloidal sample of AgNPs mediated by 1000 μ l of extract, (d) AgNPs prepared by 500 μ l extract, and (e) AgNPs synthesized by 250 μ l of extract.

215x227mm (150 x 150 DPI)

Figure 4. (upper panel) Different levels during the formation of AgNPs using the aqueous extract of *C. colocynthis* (a) transparent precursor solution, (b) solution after mixing the extract by stirring and heating, a slight color change to light yellowish brown, (c) further evident color change with time during heating and stirring indicate the reduction of Ag-ions to Ag-atoms and formation of AgNPs (d) completion of the reaction with reddish brown color of AgNPs. (lower panel) (a) AgNO₃ colorless precursor solution, (b) light yellow color *C. colocynthis* aqueous extract, (c-d) colloidal sample of AgNPs prepared by 4ml, 2.5 ml and 1 ml of *C. colocynthis* extract respectively.

254x193mm (150 x 150 DPI)

Figure 5. Schematic illustration of synthesis mechanism of extract mediated AgNPs. The phenomena occurring in each phase is mentioned accordingly.

286x153mm (150 x 150 DPI)

Figure 6. UV-Vis spectroscopy results of AgNPs synthesized by using; (a) three different concentrations of *A. indica* leaves extract, (b) three different concentration of *C. colocynthis* fruit aqueous extract. Single absorption peak in each case appearing in the wavelength range of 412 nm to 430 nm indicate the spherical shaped nano-sized silver particles. Insets display images of each colloidal sample in cuvette.

474x188mm (150 x 150 DPI)

Figure 7. SEM micrographs and size distribution histograms of both types of biosynthesized AgNPs; (a) SEM image of *A. indica* mediated AgNPs showing the almost spherical shaped NPs decorated on the larger sized extract particles, (b) particle size histogram of *A. indica* AgNPs indicating the diameter range 12-24 nm. (c) SEM results of *C. colocynthis* mediated AgNPs revealing the sphere like morphology and nice random dispersal over the bigger extract particles. The average size of these AgNPs was found to be 26 ± 5 nm as can be seen in the histograms (d).

237x187mm (150 x 150 DPI)

Figure 8. EDX spectrum showing elemental composition analysis of AgNPs prepared by the aqueous extracts of (a) *A. indica* leaves and (b) *C. colocynthis* fruit. Insets show the table of elements in each case. The presence of elemental silver and other biomaterials can be discerned by different absorption characteristics peaks in the both spectra.

266x313mm (150 x 150 DPI)

Figure 9. X-ray diffraction patterns of (a) *A. indica* mediated AgNPs and (b) *C. colocynthis* mediated AgNPs. In both cases, along with characteristics peaks designating the face centered cubic (fcc) crystalline nature of prepared nanoparticles, appearance of few extra peaks (mentioned by stars *) indicate the presence of some extract molecules.

476x190mm (150 x 150 DPI)

Figure 10. FTIR spectra of both types of AgNPs; (a) *A. indica* leaves extract mediated AgNPs, (b) AgNPs prepared using extract of *C. colocynthis* fruit. Occurrence of various absorption peaks at different positions in each case indicates the presence of difference biomolecules on the surface of synthesised nanoparticles.

376x152mm (150 x 150 DPI)

Figure 11. A representative display of larvicidal bioassay for different test samples in the beakers including water control (water), *A. indica* extract and its mediated AgNPs at different concentrations (as mentioned); upper panel shows the initial stage of experiment with 0h time while results after 24 h are represented in lower panel. It can be noticed that in the beginning all larvae were active and randomly scattered but after 24h due to mortality most of the dead larvae are gathered in the middle of the beakers. However, in control, extract and samples with low AgNPs concentration scattered larvae can be discerned.

370x118mm (150 x 150 DPI)

A typical demonstration of larvicidal activity of *C. colocynthis* extract mediated AgNPs with different concentrations, as prepared extract and water as control. In upper panel, different samples with larvae are shown in the start of experiment (at 0h) where random sprinkle of the larvae into the liquid can be seen. After 24h of exposure, larvicidal efficacy of different samples is presented in the lower panel where dead larvae can be found assembled at the bottom of the beakers.

370x118mm (150 x 150 DPI)

Figure 13. Graphs showing the larvicidal performance in mortality (%) of both types of AgNPs at different test concentrations; (A) *A. indica* extract mediated AgNPs showed LC50 at about 1.25 ppm while at 20 ppm 100% mortality was observed, (B) AgNPs prepared by *C.colocythis* extract exhibited 50% mortality (LC50) only at about 0.3 ppm and 100% mortality at 5ppm.

283x74mm (150 x 150 DPI)

Appendix B

Reply to Reviewer's Reports

(Manuscript ID RSOS-200540)

We are thankful to the reviewers for their expert assessment of our manuscript. We are grateful for their thoughtful and valuable comments to improve our manuscript. All points and issues raised by the learnt reviewers have been considered and the manuscript has been modified accordingly. A detailed point-by-point response to all comments is provided below indicating the implemented changes in in the revised version of the manuscript. A highlighted version by 'Track Changes' is also included with the resubmission.

Response to 1st Reviewer's Comments

Comment 1. The article describes the synthesis of Ag nanoparticles using *Azadirachta indica* and *Citrullus colocynthis* extracts. The work done is useful and is in line with the current scientific demands. There are also some issues to be addressed before publishing this work.

Response 1. We are grateful to the learnt reviewere for mentioning our work "*useful*" and "*in line with current scientific demands*".

Comment 2. There are several grammatical mistakes for instance I am highlighting some of these:

- i) page 2 line 17 which have famous is incorrect it should be which are famous
- ii) page 2 line 25 "vector for many deceases like" it should be vector for many diseases like
- iii)page 2 line 25 please correct "fever.Dengue"
- iv) page 2 line 35 "AgNPs were manufactured" were synthesized
- v) page 2 line 35 "the mixture was let to cool down to room temperature" should be the mixture was cooled to room temperature

The authors should thoroughly check the whole manuscript and English should be improved as well as all the editorial mistakes should be rectified.

Response 2. We are thankful to the reviewer for pointing out these grammatical mistakes to improve beauty of the text. The whole manuscript has been thoroughly checked and all the grammatical and editorial mistakes mentioned by the reviewers and otherwise are corrected.

Comment 3. Page 6 line 10, authors mentioned the particle size, I guess it should be crystallite size?

Response 3. The corrections were made as suggested by the reviewer.

Comment 4. Authors have not mapped the particle size in SEM images. The particle size should be measured in SEM image and it should be visible in the picture.

Response 4. According to the reviewer recommendations, the SEM images have been modified by adding the new images showing particle size visible in the picture.

Comment 5. The authors have mentioned that the chosen plants have known biological activity. However, they do not show any activity against *A. aegypti*. Secondly, if the extracts are not active against *A. aegypti* why there is a large difference in activity of AgNPs prepared from two different extracts as LC50 at 1.25 ppm for *A. indica* and LC50 at 0.3 ppm for *C. colocynthis* is observed. The later is 4 times more active than the former one. I do not seem that activity is only due to AgNPs in that case it should be similar or close to each other.

Response 5. We are appreciative to the reviewer for highlighting this point. Firstly, it is well-established fact that various medicinal plants including *A. indica* and *C. colocynthis*, exhibit different biological activities (R1-R3). However, in our case, both of the aqueous extracts (*A. indica* leaves and *C. colocynthis* fruit) did not show any larvicidal efficacy against dengue vector. The inefficiency of aqueous extracts of different plants was also reported by other researchers [R3-R6] however, some studies report concentration-dependent biological efficiency of aqueous extracts of some natural products [R2,R7]. Thus, biological activity of extracts of biomaterials depend on various

factors such as nature, different parts (leave, fruit, stem and seed) and level of purification (crude, highly purified, aqueous, or alcoholic) and concentration [R8-R9].

The reason of the difference in larvicidal activity of AgNPs prepared by *A. indica* leaves or *C. colocynthis* fruit is that different types of biomolecules are deposited on the surface of AgNPs (as confirmed by FTIR analysis). This is, in fact, the synergistic effect of Ag-ions and biomolecules to enhance the larvicidal performance of extract mediated AgNPs. Thus the difference in larvicidal efficiency occurs due to the presence of different types of biomolecules on the surfaces of AgNPs although Ag-ions are same in both cases. Therefore, the LC₅₀ values for both cases were different (LC₅₀ at 1.25 ppm for *A. indica* and LC₅₀ at 0.3 ppm for *C. colocynthis*).

Main text has also been modified to explain this point.

Comment 6. Figures 1, 2 and 5 are unnecessary and should be deleted.

Response 6. Figures 1 and 3 and figures 2 and 4 are combined according to the recommendations of reviewer 2, instead of deleting.

Figure 5 depicts schematics of the potential formation mechanism of extract mediated AgNPs and different phases of nucleation and growth processes. We would like to keep it in the main text with the kind consent of our proficient reviewer.

Data of all modified figures has been uploaded to Dryad Digital Repository

Our data are deposited at Dryad:

<https://datadryad.org/stash/dataset/doi:10.5061/dryad.z612jm68k> [

Dryad Reviewer URL:

<https://datadryad.org/stash/share/DksW82FCDqISwPqI2aYhDI4C9QUE74hEBPTmrJ3trHw>

Comment 7. the sentence in conclusion section “Both types of NPs were found pure metallic silver in nature having FCC crystalline structure and of spherical in shapes with diameters in the range of 25±5 nm, synthesized by different biomolecules of extracts as determined by different characterization techniques including SEM, UV-Vis, XRD and FTIR” is too long and understandable. It should be refabricated to two sentences.

After addressing these issues the article can be published

in RSOS.

Response 7. We are agreed with the reviewer and this sentence has been rephrased into two sentences as,

“Both types of NPs were found pure metallic silver in nature having FCC crystalline structure and were spherical in shapes with diameters in the range of 25 ± 5 nm, as determined by SEM and XRD. The optical behavior of colloidal samples and presence of different extract biomolecules were studied by UV-Vis, and FTIR.” The main text has also been modified accordingly

We are obliged to the reviewer for recommending our manuscript for “*publication*” in RSOS.

Response to 2nd Reviewer’s Comments

Comment 1. The MS presents work which has been published with various other plants. The work carried out is significant but need major revision.

Response 1. We are obliged to the reviewer for mentioning our work “*significant*”.

Comment 2. MS needs enormous revision in scientific reporting, language and grammar. The scientific names are wrongly spelled at some places.

Response 2. We are grateful to the learnt reviewer for highlighting the errors to enhance the flouncy and beauty of the text. The manuscript is carefully examined and all mistakes of language, grammar, typos and scientific reporting including scientific names pointed out by the reviewer and otherwise are corrected and highlighted by ‘Track Changes’ in the revised manuscript.

Comment 3. Each section of the MS needs major revision. The comments have been marked on the MS. Authors are advised to go through each and every comment; and rectify accordingly.

Response 3. We are thankful to the reviewer for valuable comments and suggestions. We have addressed all the comments and points raised by the learnt reviewer and

manuscript has been thoroughly revised in the light of remarks and recommendations of the reviewer.

The detailed response to the comments is described below to highlight the amendments made in the revised version of the manuscript accordingly.

Comment: The MS is about anti-dengue vector. Kindly revise.

Response : The title of the MS has been rectified as directed by the reviewer.

Comment: Check series.... FTIR was done in last..Please write based on conducting the characterization

Response : In abstract, series of the characterizations techniques has been corrected, as suggested by the reviewer.

Comment: Authors have not added any comparison between the characteristics of two kinds of NPs. Some features may be mentioned.

Response : We agreed to the reviewer and the abstract has been modified by adding comparative characteristic features of both types of AgNPs.

Comment: write scientific names correctly throughout the manuscript. Start specific names with small case. Take care of spellings. *A. indica* and *C. colocynthis*

Response: All scientific names and terms have been corrected including *A. indica* and *C. colocynthis*, as directed by the reviewer.

Comment: Which part? As you have prepared NPs with fruits, please specify

Response : The *C. colocynthis* fruit has been specified and reference has also been added in the main text, as suggested by the reviewer.

Comment: Dengue is transmitted by *Aedes* mosquitoes. It is caused by virus. Larvae have no role in it. Please rectify Write as hyphenated throughout the MS

Response: We agreed to the reviewer and the sentence has been rectified in the main text as; “Dengue fever incidence has increased 30-fold worldwide in the last 30 years which is caused by dengue virus”

Comment: What do you mean by glass substrate?

Response : Glass substrate means just a small piece of glass slide on which few drops of colloidal AgNPs sample are dried to make a thick enough NPs layer. This was used as sample for characterization purposes. It has also been described in literature [R10, R11].

Comment: 1ml, 2.5 ml and 4ml..... on what basis the ratios (pattern of conc.) were decided? The amount/conc. of a compound to be used in a mixture is always in a particular ratio. For example...1, 2, 4, 8....

However, only AgNPs prepared by using the 500 µl of extract was used in larvicidal activity and other characterization, specify the reason plz

Response: In this study, two types of extracts were used as reducing and stabilizing agents for synthesis of AgNP. The amount of extracts in both cases was selected from the literature protocols [R12, R13]. To obtain the optimized properties three different amounts of each extract were used for the preparation of AgNPs. An amount was selected and then its 2 fold and 4 fold was used for AgNPs synthesis process. In the case of *A. indica* extract, 250 µL, 500 µL and 1000µL while for *C. colocynthis* extract 1 mL, 2 mL and 4 mL were used (2.5 mL was typo error which has been corrected in the main text). The AgNPs prepared from 500 µL (*A. indica*) and 2 mL (*C. colocynthis*) were taken as optimized AgNPs on the basis of Uv-Vis results. This point has been addressed in the main text according to the reviewer’s recommendations.

Comment: Why did authors use directly 5 mM concentration to prepare NPs? Higher concentration of silver nitrate itself is toxic? Did you try for lower concentrations? What was the result? If not, then why?

Response: We agreed to the reviewer that AgNPs can be made by different concentrations of precursor AgNO₃ solution (Lower and higher than 5mM). In this study, both types of AgNPs were prepared following the protocols mentioned in the literature. *A. indica* extract-mediated AgNPs were synthesized using 1mM solution of AgNO₃ using methods described by of Aparajita et al. [R12] while *C. colocynthis* extract-mediated AgNPs using 5mM concentration of precursor solution as reported by Shawkey et al. [R13].

Comment: We cannot say that the fruit extract was more effective at lower concentration as the comparison is between 1 mM and 5 mM of AgNO₃. 5 mM is itself a very high concentration of silver nitrate. If authors want to compare the efficacy then they should do at similar concentrations either 1 mM or 5 mM

Response: In our work we used two different types of extracts and AgNPs to study the larvicidal activity. Each type of AgNPs were prepared following different protocols using different precursor concentration (1mM or 5mM) and two different reducing agents (*A. indica leaves* extract or *C. colocynthis fruit* extract). We aim to report the antilarvae performance of each type of extract and AgNPs at different concentrations. It is well-known fact that change in concentration of precursor solution will affect the size and morphology of prepared AgNPs [R14, R15]. In our case, we used two type of extracts as reducing and stabilizing agents and thus different types of biomolecules were deposited on the surface of each type of AgNPs which caused variation in larvicidal efficacy.

We have rectified the main text (abstract, result and discussion and conclusions) to address the comparison issue raised by the reviewers.

Comment: ppm unit is no longer in use. May be replaced

Response: According to reviewer recommendations, ppm has been replaced by mg/L throughout the manuscript.

Comment: “The test concentrations of both types of AgNPs were selected on the basis of pilot experiments in which LC50

concentrations were found”.....Any published result? If so, give ref. Otherwise, you may give values obtained here

Response: The LC₅₀ values of pilot experiments were found to be 1.25 mg/L and 0.3 mg/L for *A. indica* and *C. colocynthis* mediated AgNPs respectively. These LC₅₀ values have also been mentioned in the main text according to the reviewer suggestions .

Comment: This time dependent color change during the synthesis, actually,time not mentioned

Response: Time has been mentioned in the main text as “10 min for *A. indica* mediated AgNPs and 20 min for *C. colocynthis* mediated AgNPs.”

Comment: we can notice strong silver signal along with some other signals. mention Ag%

Response: % age values of Ag have been mentioned for both types of AgNPs in main text as “37.08 % for *A. indica* AgNPs and 38.5% for *C. colocynthis* AgNPs” as directed by the reviewer.

Comment: mL denotes the volume. As per WHO protocol, 1 mL of toxicant is added to 249 mL of water to study larval mortality. What about concentration? Write the conc of extract. How can you compare the mL of extract with ppm of AgNPs?

Response: The reviewer is right that ml determines the volume only. We have added percentage concentration of the extract. It is in common practice to use the extract in percentage concentration [R16, R17]. The prepared extract is standardized as 100 % from which dilutions are drawn for biassays. We used 3 different amounts (1mL, 2mL and 3mL) of the extract to be dissolved in water to make a final volume of 200 mL. 1 mL of extract in 200 mL of water will make a 0.5 % extract concentration, 2 mL will make 1 % and 3 mL will make 1.5 % extract concentration in the solution. Furthermore because the dynamics of the concentration formulations for extract and nanoparticles is different, extract is based on the extracted material while nanoparticles' concentration is

devised from the precursor used, therefore, it is not necessary to have a direct comparison of the two in terms of concentration units.

Comment: 1ml, 2ml, 4 ml..... Are these concentrations? , 2, 3 mL.... Is this concentration?

Response: No, these are quantities, now we have added % age concentration of extracts (0.5 %, 1%, and 1.5%) in main text to rectify this issue raised by the reviewer.

Comment: How can you make anti-dengue medicine which has been tested against dengue larvae not dengue virus

Response: We are thankful to the reviewer for indicating this point, we have rectified the main main text as “This indicates that antilarval pharmaceuticals can be formulated using very low concentration of natural products based AgNPs for practical applications”

Comment: Figure captions: all figure captions need to be revised...as they too lengthy, unnecessarily too big legends. Please make all legends concise and crisp.

Response: All the figure captions have been revised to make brief and concise, as directed by reviewer.

Comment: Figs. 1, 2 and 3 can be easily combined. Please shorten and make it crisper. F1: explanations can be included in the figures itself (labellings of figures)

Response: According to the reviewer recommendations, previous figure 1 and 3 have been combined into one (new figure no. 1) and figure 2 has been combined with figure 4 to make new figure no. 2. The main text has also been amended accordingly.

Comment: F11 and F12: Please delete...instead add LC values tables

Response: The LC values have already been added in tables 1 and 2. The figures 11 and 12 provide the visual demonstration of the larvicidal activity, so with the kind consent of our proficient reviewer we would like to keep them in the main text.

References:

- R1. Abu-Darwish MS, Efferth T. 2018 Medicinal plants from near east for cancer therapy. *Front. Pharmacol.* 9, 56. (doi: 10.3389/fphar.2018.00056)
- R2. Kadir SLA, Yaakob H, Mohamed RZ. 2013 Potential anti-dengue medicinal plants: a review. *J Nat Med* 67:677–689 (doi: 10.1007/s11418-013-0767-y)
- R3. Govindachari TR, Suresh G, Gopalakrishnan G, Banumathy B, Masilamani S. 1998 Identification of antifungal compounds from the seed oil of *Azadirachta indica*. *Phytoparasitica* 26, 109–116. (doi: 10.1007/BF02980677)
- R4. Shawkey AM, Abdulall AK, Rabeh MA, Abdellatif AO. 2014 Enhanced biocidal activities of *Citrullus colocynthis* aqueous extracts by green nanotechnology. *International Journal of Applied Research in Natural Products* 7 (2), 1-10.
- R5. Shawkey AM, Rabeh MA, Abdulall AK, Abdellatif AO. 2013 Green nanotechnology: Anticancer activity of silver nanoparticles using *Citrullus colocynthis* aqueous extracts. *Adv. Life Sci. Technol.* 13, 60–70.
- R6. Rawani A, Ghosh A, Chandra G. 2013 Mosquito larvicidal and antimicrobial activity of synthesized nano-crystalline silver particles using leaves and green berry extract of *Solanum nigrum* L. (Solanaceae: Solanales). *Acta Tropica* 128, 613–622. (doi: 10.1016/j.actatropica.2013.09.00)
- R7. Sujitha V, Murugan K, Paulpandi M, Panneerselvam C, Suresh U, Roni M, Nicoletti M, Higuchi A, Madhiyazhagan P, Subramaniam J, Dinesh D, Vadivalagan C, Chandramohan B, Alarfaj AA, Munusamy MA, Barnard DR, Benelli G. 2015 Green-synthesized silver nanoparticles as a novel control tool against dengue virus (DEN-2) and its primary vector *Aedes aegypti*. *Parasitol Res.*, 114, 3315-25. (doi: 10.1007/s00436-015-4556-2)
- R8. Suganya G, Karthi S, Shivakumar MS. 2014 Larvicidal potential of silver nanoparticles synthesized from *Leucas aspera* leaf extracts against dengue vector *Aedes aegypti*. *Parasitol Res.* 113, 875–880. (doi: 10.1007/s00436-013-3718-3)
- R9. Suresh U, Murugan K, Benelli G, Nicoletti M, Barnard DR, Panneerselvam C, Kumar PM, Subramaniam J, Dinesh D, Chandramohan B. 2015 Tackling the growing threat of dengue: *Phyllanthus niruri*-mediated synthesis of silver nanoparticles and their mosquitocidal properties against the dengue vector *Aedes aegypti* (Diptera: Culicidae). *Parasitol Res.* 114(4), 1551-1562. (doi: 10.1007/s00436-015-4339-9)

- R10. Kanwal Z, Raza MA, Riaz S, Manzoor S, Tayyeb A, Sajid I, Naseem S. 2019 Synthesis and characterization of silver nanoparticle-decorated cobalt nanocomposites (Co@AgNPs) and their density-dependent antibacterial activity. *R. Soc. open sci.* **6**, 182135. (doi:10.1098/rsos.182135)
- R11. Kanwal Z, Raza, MA, Manzoor F, Riaz S, Jabeen G, Fatima S, Naseem S. 2019 A Comparative Assessment of Nanotoxicity Induced by Metal (Silver, Nickel) and Metal Oxide (Cobalt, Chromium) Nanoparticles in *Labeo rohita*. *Nanomaterials*, **9**, 309. (doi:10.3390/nano9020309)
- R12. 22-Aparajita V, Mohan SM. 2016 Controllable synthesis of silver nanoparticles using Neem leaves and their antimicrobial activity. *J Radiat Res Appl Sc* **9(1)**, 109-115. (doi: 10.1016/j.jrras.2015.11.001)
- R13. 23-Shawkey AM, Rabeh MA, Abdulall AK, Abdellatif AO. 2013 Green nanotechnology: Anticancer activity of silver nanoparticles using *Citrullus colocynthis* aqueous extracts. *Adv. Life Sci. Technol.* **13**, 60–70.
- R14. Devadiga A, Vidya SK, Saidutta MB. 2017 Effect of Precursor Salt Solution Concentration on the Size of Silver Nanoparticles Synthesized Using Aqueous Leaf Extracts of *T. catappa* and *T. grandis* Linn f.—A Green Synthesis Route. In: Mohan B. R., Srinikethan G., Meikap B. (eds) *Materials, Energy and Environment Engineering*. Springer, Singapore. (doi: 10.1007/978-981-10-2675-1_17)
- R15. Vishwakarma K, Shweta, Upadhyay N, Singh J, Liu S, Singh VP, Prasad SM, Chauhan DK, Tripathi DK, Sharma S. 2017 Differential Phytotoxic Impact of Plant Mediated Silver Nanoparticles (AgNPs) and Silver Nitrate (AgNO₃) on *Brassica sp.* *Front Plant Sci* **8**, 1501. (doi: 10.3389/fpls.2017.01501)
- R16. Subramaniam K, Siswomihardjo W, Sunarintyas S. 2005 The effect of different concentrations of Neem (*Azadiractha indica*) leaves extract on the inhibition of *Streptococcus mutans* (In vitro). *Maj Ked Gigi (Dent J)* **38**, 176-9. ..
- R17. Ederley V, Gloria C, Gladis M, César H, Jaime O, Oscar A. 2018 Silver Nanoparticles Obtained by Aqueous or Ethanolic *Aloe vera* Extracts: An Assessment of the Antibacterial Activity and Mercury Removal Capability. *J Nanomater* 2018, Article ID 7215210, 7 pages (doi: 10.1155/2018/7215210)

Appendix C**ROYAL SOCIETY
OPEN SCIENCE****Biosynthesis, Characterization and Anti-dengue vector
activity of Silver Nanoparticles prepared by Azadirachta
indica and Citrullus colocynthis**

Journal:	Royal Society Open Science
Manuscript ID	RSOS-200540.R1
Article Type:	Research
Date Submitted by the Author:	04-Jul-2020
Complete List of Authors:	Rasool, Shafqat; University of the Punjab, Quid-e-Azam Campus, Lahore-54590, Pakistan, Centre of Excellence in Solid State Physics Raza, Muhammad Akram; University of the Punjab, Centre of Excellence in Solid State Physics Manzoor, Farkhanda ; Lahore College for Women University, Department of Zoology Kanwal, Zakia; Lahore College for Women University, Jail Road, Lahore-54000,, Department of Zoology Riaz, Saira; University of the Punjab, Centre of Excellence in Solid State Physics Iqbal, Muhammad Javaid; University of the Punjab, Centre of Excellence in Solid State Physics Naseem, Shahzad ; University of the Punjab, Centre of Excellence in Solid State Physics
Subject:	Green chemistry < CHEMISTRY, Nanotechnology < CHEMISTRY, biophysics < CROSS-DISCIPLINARY SCIENCES
Keywords:	silver nanoparticles, green synthesis, Citrullus colocynthis, Azadirachta indica, antilarval activity
Subject Category:	Chemistry

Author-supplied statements

Relevant information will appear here if provided.

Ethics

Does your article include research that required ethical approval or permits?:

This article does not present research with ethical considerations

Statement (if applicable):

CUST_IF_YES_ETHICS :No data available.

Data

It is a condition of publication that data, code and materials supporting your paper are made publicly available. Does your paper present new data?:

Yes

Statement (if applicable):

Data Accessibility statement:

Our data are deposited at Dryad:

<https://datadryad.org/stash/dataset/doi:10.5061/dryad.z612jm68k> [48], with following reviewer URL;

Dryad Reviewer URL:

<https://datadryad.org/stash/share/DksW82FCDqISwPqI2aYhDI4C9QUE74hEBPTmrJ3trHw>

Conflict of interest

I/We declare we have no competing interests

Statement (if applicable):

CUST_STATE_CONFLICT :No data available.

Authors' contributions

This paper has multiple authors and our individual contributions were as below

Statement (if applicable):

Authors' Contributions:

- 1). SR and MAR performed the synthesis experiments, analysed the data and wrote the manuscript
- 2). ZK, SR and MAR designed the experiments and conducted the larvicidal bioassays
- 3). SR and SN conducted the characterization measurements and helped in interpretation of data
- 4). FM and ZK provided the larvae facility and helped in larvicidal activity experiments
- 5). SR and SN conducted the characterization measurements and helped in interpretation of data
- 6). MJR and SR conducted the FTIR and SEM characterization and helped in data analysis

Biosynthesis, Characterization and Anti-dengue **vector** activity of Silver Nanoparticles prepared by *Azadirachta indica* and *Citrullus colocynthis*

Shafqat Rasool¹, Muhammad Akram Raza^{*1}, Farkhanda Manzoor², Zakia Kkanwal², Saira Riaz¹, Muhammad Javaid Iqbal¹, Shahzad Naseem¹

¹Centre of Excellence in Solid State Physics, University of the Punjab, Quid-e-Azam Campus, Lahore 54590, Pakistan

²Department of Zoology, Lahore College for Women University, Jail Road, Lahore 54000, Pakistan

Keywords: anti-larval-dengue activity, silver nanoparticles, green synthesis, *Azadirachta indica*, *Citrullus colocynthis*

Abstract

We report here biosynthesis of silver nanoparticles (AgNPs) using aqueous extracts of (i) *Azadirachta indica* leaves and (ii) *Citrullus colocynthis* fruit and their larvicidal activity against *Aedes aegypti*. The UV-Vis spectroscopy absorption peaks occurred in the range of 412-416 nm for *A. indica* AgNPs and 416-431 nm for *C. colocynthis* AgNPs indicating the silver nature of prepared colloidal samples. The Scanning electron microscopy examination revealed the spherical morphology of both types of NPs with average size of 17±4 nm (*A. indica* AgNPs) and 26±5 nm (*C. colocynthis* AgNPs). The x-ray diffraction pattern confirmed the face centred cubic (FCC) structure with crystallite size of 11±1 nm (*A. indica* AgNPs) and 15±1 nm (*C. colocynthis* AgNPs) while characteristic peaks appearing in Fourier transform infrared spectroscopy analysis indicated the attachment of different biomolecules on AgNPs. The larvicidal activity at different concentrations of synthesized AgNPs (1-20 mg/L) and extracts (0.5-1.5 %) against *Aedes aegypti* was examined for 24 h. A concentration dependent larvicidal potential of both types of AgNPs was observed. The LC₅₀ values were found to be 0.3 mg/L and 1.25 mg/L for *C. colocynthis* AgNPs and *A. indica* AgNPs respectively. However, both extracts did not exhibit any notable larvicidal activity.

1. Introduction

Nanotechnology, owing to unique, extraordinary and incredible properties of materials at nano-scale, has revolutionized almost every field of science and technology. Especially, in arena of medical science and medicine, it is considered as next logical step and the future medicine [1, 2]. However, the potential risks such as harmful side effects on human and environment has made nanotechnology a double-edged sword [3,4].

Natural products based synthesis of nanoparticles can help to reduce the hazards of nanotechnology and thus can boost its applications. Biosynthesis, based on green chemistry principles, is proactive approach and has numerous advantages over conventional chemical and physical methods such as cost-effectiveness, environment friendly, simple and safe, no use of hazardous chemicals as reducing agents, less wastage of materials, minimum energy usage, safer disposal and recycling [5, 6]. Studies are also reported on the preparation of various types of nanostructures by using natural products such as vitamins, biodegradable polymers, enzymes, polysaccharides and different microorganism including algae, fungi, bacteria and viruses. Nevertheless, plants extract-mediated synthesis of nanomaterials is considered advantageous due to stability of nanostructures, cost efficiency, availability of plants, lower contamination risks, require little maintenance, and simple to scale up as plants are nature's 'chemical factories' [6,7].

Silver Nanoparticles are famous for their anti-bacterial, anti-dengue-viral and anti-fungal activities [8]. AgNPs are also used in textile industry, food packing, cosmetics, paints and detergents owing to their anti-microbial activity [9]. Preparation of AgNPs can be carried out by using extract of different parts of plant such as roots, stem, leaves or fruit. In the synthesis of AgNPs by plants, polyphenols play the main role in degradation of different organics compounds. These plant extracts usually play the dual role in the synthesis process as reducing and capping agent. During green synthesis, the coatings of various biomolecules on surfaces of AgNPs not only improve their stability but also enhance their biocompatibility by reducing the toxicity risks [8, 9].

In this study, we choose *Azadirachta Azedarachta Indica-indica* (*A. indica Indica*) and *Citrullus Colocynthis colocynthis* (*C. Colocynthiscolocynthis*) for the synthesis of AgNPs because both of these plants are famous for good medicinal properties and exhibit effective biological activities. *A. indica Indica*-plant belongs to family of *Meliaceae* and has enormous uses in medical field from ancient times [10]. It is familiar as anti-bacterial, anti-fungal and anti-microbial plant as it has quercetin, β-sitosterol and nimbinin in their leaves and azadirachtin in its seeds [11]. The *A. indica Indica*

*Author for correspondence (Muhammad Akram Raza; Akramraza.cssp@pu.edu.pk).

†Present address: Centre of Excellence in Solid State Physics, University of the Punjab, Lahore-54590, Pakistan

plant extract has been reported to increase antimicrobial activity of extract-mediated nanoparticles [10, 11]. *C. Colocynthis-colocynthis* (also known as bitter apple) is a member of *Cucurbitaceae* family. The *C. colocynthis* fruit contains many biomolecules including alkaloids, glycosides, flavonoids, and fatty acids which have are famous for important biological activities such as antimicrobial, antioxidants, cytotoxic, antidiabetic, antilipidemic, and insecticide [12]. That's why it is considered one of the best medicinal plants and is used in different biomedical applications including anti-microbial, anticancer activities [12-14].

According to World Health Organization (WHO), mosquitoes can be listed as one of the deadliest organisms because millions of humans die due to different diseases caused spread by them. Mosquitoes are the potential vector for many diseases like dengue, malaria, west Nile, chikungunya, Zika and yellow fever. Dengue fever incidence has increased 30-fold worldwide in the last 30 years which is caused by *A. aegypti* mosquito vector. Dengue incidence has increased 30-fold worldwide in the last 30 years which is caused by dengue mosquito larvae; *Aedes aegypti* (*A. aegypti*) [15]. Treatment of these diseases is a challenge. To control the rapidly increasing mosquito-borne risks, researchers are working on different strategies. Since mosquitoes breed in water and at larval stage, it is easy to target them there. However, inhibiting larvae in water by conventional ways using pesticides can increase the toxic risks for environment and humans. Thus natural pesticides such as plant extracts can be simple and side effect free promising methodologies. The use of green chemistry based nanomaterials can further enhance the effectiveness and efficiency of such approaches [16-19].

In this work, *A. indica* and *C. Colocynthis-colocynthis* mediated AgNPs were manufactured synthesized, characterized and tested for their anti-dengue larvicidal activity against *A. aegypti* and it was found that these AgNPs can potentially be considered good anti-larvicidal agents.

2. Materials and Methods

2.1. Material and chemicals

Fresh leaves of *A. indica* were collected from the Botanical garden of University of the Punjab, Lahore, Pakistan and *C. colocynthis* fruit was obtained from local market of Lahore, Punjab, Pakistan. Both products were identified by the experts of Department of Botany, University of the Punjab, Lahore. Late fourth instar larvae of *A. aegypti* were collected from local pond and identified by the expert of the Department of Zoology, Lahore College for Women University, Lahore, Pakistan. Silver nitrate (AgNO_3) was of research grade and obtained from Merck (Germany). To prepare synthesis solutions, aqueous extracts and for all other purposes deionized (DI) water was used throughout the experiment.

2.2. Preparation of *A. indica* and *C. colocynthis* extracts

The aqueous extract of *A. indica* leaves was prepared following the method reported by Pragyan et al [20] with some modifications. To remove any dust and contaminations, first fresh leaves were rinsed carefully with running tap water then washed twice with deionized (DI) water and let them dry were dried in air. 20 g of these leaves were taken and converted cut into very small pieces by cutting then which were transferred to 250 mL conical flask of 250 mL and boiled for 10 min in 100 mL deionized DI water. After boiling, the mixture was let to cool down cooled to room temperature and finally filtered with Whatman No. 1 filter paper before storing at 4 °C for next step. Different steps of the preparation of *A. indica* leaves aqueous extract are shown in upper panel of Figure 1. One can notice that greenish color of mixture solution (figure 1c, before boiling) changed to yellowish light brown color in final stage (figure 1d).

The aqueous extract of *C. colocynthis* fruit was prepared by the technique mentioned elsewhere [21] with few changes and whole process of extract preparation is demonstrated in upper panel of figure 2. Briefly, a greenish white *C. colocynthis* fruit (as shown in figure 2a) was cleaned by washing many times with fresh tap water to get rid of any debris or any other contaminations, then rinsed two times with DI water before cutting into small pieces. 50g pieces of *C. colocynthis* were soaked in 200 mL of deionized DI water for 72 h at room temperature. Finally mixture was filtered 2 times with Whatman No.1 filter paper to get greenish yellow color of the extract (figure 2c) and kept at 4 °C for further use.

2.3. Synthesis of extract-mediated silver nanoparticles

To prepare AgNPs using aqueous extracts, first precursor stock solution (100mM) was made by dissolving 1.69 g of silver nitrate (AgNO_3) into 100 mL of DI water.

A. indica extract-mediated AgNPs were made by the protocol of Aparajita et al. [22] with some amendments. Three different amounts of prepared We, in our case, used three different amounts of as-prepared extract (250 μL , 500 μL and 1000 μL) were used for the same amount 20 mL of silver nitrate precursor solution (1mM) to obtain the optimized concentration. In figure 3-1 (upper-middle panel), different phases of AgNPs preparation using the *A. indica* extract are presented. First of all colorless precursor solution of AgNO_3 was heated to 70 °C with gentle magnetic stirring of at 200 rpm (figure 3a1e). Then *A. indica* extract was added dropwise into the solution and the mixture was kept for heating at hot plate at 70 °C under continuous stirring for 10 min (figure 3b1f). The color of the solution changed with time from transparent to light yellow to yellowish brown indicating the formation of AgNPs (figure 3e1g). At the end, the solution was let to cool down cooled to room temperature and kept at -4 °C for further use. Depending upon the amount of the extract (reducing agents) the color of final stage sample (colloidal AgNPs) was found to changed as can be noticed in lower panel of figure 3-1, (lower panel), from yellowish brown to reddish brown to reddish dark brown for 250 μL , 500

For the synthesis of AgNPs using aqueous extract of *C. colocynthis*, approach described by Shawkey et al. [23] was followed with some alterations. Briefly, First 20 mL of 5mM silver nitrate solution was mixed with different amounts (1 mL, 2 mL and 4 mL) a defined amount of *C. colocynthis* extract and then the mixture was heated to the boiling temperature (about 95 °C) under constant stirring of at 200 rpm for 20 min. Finally, solution was cooled down to room temperature and store at -4 °C. Color changes during the synthesis process indicate different phases of the synthesis and are presented in figure 4-2 (upper-middle panel). We synthesized AgNPs using three amounts of as-prepared extract of *C. colocynthis* viz., 1ml, 2.5 ml and 4 ml following aforementioned method to achieve the optimized concentration. All three colloidal AgNPs samples mediated by different amounts of extract with precursor and extract samples are shown in figure 4-2 (lower panel). Nevertheless, only 2.5 mL extract-mediated AgNPs were selected as optimized AgNPs based on Uv-Vis results further to used for larvicidal activity and other characterizations.

[revised manuscript text omitted]
 on the surface of AgNPs, in both cases, which provided the required bio-reduction and efficient stabilization during the synthesis process, FTIR measurement were carried out and obtained spectra are presented in figure 10.~~

FTIR spectrum of AgNPs synthesized from *A. indica* leaves is shown in figure 10a-8a where various absorption bands at different specific frequency indicate a composite nature. The prominent peaks occurred at the frequency ranges of 3265, 2919, 2849, 2363, 1734, 1616, 1244, 1161, 1097 and 1027 cm⁻¹ as can be noticed from the figure 10a8a. A broad band appearing in the range of 3265 cm⁻¹ can be attributed to the stretching vibrations of O-H bond indicating the presence of polyphenols. The peaks occurring at value range of 2919 cm⁻¹ and 2849 cm⁻¹ can be assigned to C=C aromatic and C-H alkaline (asymmetric stretching of C-H bonds) groups [40]. The possible presence of C≡N and C≡C triple bonds can be ascribed by peaks of 2363 cm⁻¹ because band appearing in the range of 2200-2400 cm⁻¹ indicate the C-N and C-C triples bonds. The band at 1734 cm⁻¹ could be due to C=O stretching vibration in the carbonyl groups from terpenoids and flavonoids. The peaks of 1616 cm⁻¹ might indicate presence of C=C bonds or aromatic rings attributed to carbonyl stretch in proteins [41]. The absorption bands appearing at 1244, 1161, 1097 and 1027 cm⁻¹ can be

assigned to the C-H alkene (Aliphatic amines, C-H Wag, C-N stretching in amide functional group) and C-O vibrations of ether linkages [14, 40, 42]. Thus FTIR analysis of *A. indica* mediated AgNPs confirmed the presence of different phytochemical components such as flavonoids, polyphenols and terpenoids of *A. indica* extract which potentially played a vital role in the reduction of Ag⁺ ions to Ag⁰ atoms. Especially the proteins (due to the carbonyl and NH₂ of amino acid) present in *A. indica* leaves extract served as reducing and encapsulating agent because carbonyl group can reside on the metallic silver due to strong affinity for metals and thus developing a capping layer on surfaces of AgNPs [20, 36, 40, 42].

For the AgNPs prepared by *C. colocynthis*, FTIR spectrum exhibited peaks at 3255, 2920, 2361, 2341, 1634, 1582, 1404, 1339, 1240, 1141, 1095, 1007 and 953 cm⁻¹ as shown in figure 10b8b. The occurrence of absorption bands at 953-1007 cm⁻¹ might be due to C-O- or C-O-C- functional groups while the peaks at 1010-1150 cm⁻¹ can be allocated to C-N stretching vibrations related to aliphatic amines or to alcohols or phenols indicating the existence of polyphenols [14, 43]. The bands at 1240 cm⁻¹ and 1339-1460 cm⁻¹ attributed to the amide III and II groups respectively. Likewise the absorption in this region (around 1384 cm⁻¹) indicated the residual amount of NO₃. Whereas the prominent absorption band at 1582 cm⁻¹ might be owing to symmetric stretching vibrations of -COO- groups of amino acid confirming the presence of *C. colocynthis* extract protein. The existence of proteins was further indicated by the peak at 1634 cm⁻¹ due to stretch vibration of -C = C- of amide I bonds of proteins. The bands arising at around 2200-2400 cm⁻¹ indicate the probable presence of C-N or C-C triple bonds. The peaks occurring 2920 cm⁻¹ can be dispensed to C-H bond vibrations. The intense broad absorption band at 3355 cm⁻¹ can be designated to characteristic of -OH (hydroxyl) group indicating the existence of alcohols and phenolic compounds [34, 44, 45]. The FTIR results confirmed that the *C. colocynthis* mediated AgNPs were surrounded by various phytochemical constituents such as amines, aldehydes, alcohols, ketones and carboxylic acids. These *C. colocynthis* extract based biomolecules (proteins and metabolites) played dual role of bio-reduction and stabilization during the synthesis of AgNPs.

4.6. Larvicidal Bioassay

In order to study the antilarval activity of aqueous extracts of *A. indica* and *C. colocynthis* and their respective AgNPs, different concentrations were tested for larvicidal bioassay against fourth instars larvae of *A. aegypti*. In order to study the antilarval activity of prepared samples; aqueous extracts of *A. indica* and *C. colocynthis* and AgNPs mediated by these extracts at different concentrations, were subjected separately to larvicidal bioassay on fourth instars larvae of dengue (*A. aegypti*). After 24h treatment, dead and alive mosquito larvae were counted to calculate percent mortality. Larvae which showed no motility after being disturbed with a needle were considered dead. The typical demonstration of larvicidal bioassay are exhibited in figure 11-9 (*A. indica* panel) and figure 12-10 (*C. colocynthis* panel).

For *A. indica* leaves aqueous extract, three concentrations; 1 ml, 2 ml and 3 ml 0.5 %, 1 % and 1.5 % of as-prepared extract were tested while five concentrations 1.25 mg/Lppm, 2.5 mg/Lppm, 5 mg/Lppm, 10 mg/Lppm and 20 mg/Lppm of *A. indica* extract-mediated AgNPs were exposed to *A. aegypti*. DI water was taken as control in each experiment. In the initial stage of this bioassay at t= 0 h, (as shown in upper panel of figure 11-9), random scattering of the larvae indicate their active movement in all the samples. After 24 h, lower panel of figure 11-9, all larvae in the control (water) were alive and moved energetically. Beakers containing extract of *A. indica* showed no mortality in all three concentration samples. Nonetheless larvae were found less motile in the sample with 3 mL concentration of extract as compared to other two samples (1 mL and 2 mL extract). Beakers containing AgNPs showed different mortalities with different concentrations. The dead larvae can be observed at the bottom or gathered in the middle of the each beaker with high concentration (lower panel of figure 11-9). AgNPs with concentration of 1.25 mg/Lppm, 2.5 mg/Lppm, 5 mg/Lppm, 10 mg/Lppm and 20 mg/Lppm caused 59%, 66%, 80% and 85% and 100% mortality respectively. All the larvicidal results of *A. indica* leaves extract and mediated AgNPs are summarized in table 12.

Table 2 Larvicidal activity of DI water (control), aqueous extract of *A. indica* leaves and extract-mediated AgNPs at different concentrations against fourth instars larvae of *A. aegypti*.

Sample type	Concentration of sample	Percentage Mortality (%)
Water (control)	-	0 (active)
as-prepared	1 ml 0.5 %	0 (active)
Aqueous extract of	2 ml 1 %	0 (active)
A. indica leaves	3 ml 1.5 %	0 (less motile)
A. indica extract-mediated AgNPs	1.25 ppm mg/L	49±4
	2.5 mg/Lppm	66±6
	5 mg/Lppm	80±4
	10 mg/Lppm	85±4
	20 ppm mg/L	100±0

A similar larvicidal activity assay was conducted for samples of *C. colocynthis* fruit extract; in this case again three extract concentration of as-prepared aqueous extract; 1 ml, 2 ml and 3 ml 0.5 %, 1 % and 1.5 % and DI water as control were taken while for extracted mediated AgNPs five concentration were chosen as 0.3 mg/Lppm, 0.6 mg/Lppm, 1.25 mg/Lppm, 2.5 mg/Lppm and 5 mg/Lppm (figure 12-10). In the beginning of the activity test (at 0 h), all larvae were alive and motive in samples (upper panel of figure 12-10). After exposure of 24 h, again no mortality in (control) and all three extract samples, however, a concentration dependent effect on motility of larvae was noticed in extract samples. The

larvae in sample with 3 mg/L extract were least motile. In the case of AgNPs samples, highest mortality was counted in the sample with maximum concentration. A mortality of 53%, 64%, 80%, 91% and 100% were recorded for samples with AgNPs concentration of 0.3 mg/Lppm, 0.6 mg/Lppm, 1.25 mg/Lppm, 2.5 mg/Lppm and 5 mg/Lppm respectively. Assembly of immotile (dead) larvae can be witnessed in middle of beakers (lower panel, figure 1210). The statistics of *C. colocythis* mediated larvicidal assay are listed in table 23.

Table 3 Larvicidal results of DI water (control), aqueous extract of *C. colocythis* fruit and extract-mediated AgNPs at different concentrations against fourth instars larvae of *A. aegypti*.

Sample Type	Concentration of sample	Percentage Mortality (%)
Water	-	0 (active)
as-prepared aqueous extract of C. colocythis fruit	1 ml 0.5 % 2 ml 1 % 3 ml 1.5%	0 (active) 0 (less motile) 0 (least motile)
C. colocythis extract-mediated AgNPs	0.3 mg/Lppm 0.6 mg/Lppm 1.25 mg/Lppm 2.5 mg/Lppm 5 ppmmg/L	53±4 64±6 80±3 91±3 100±0

We did not observe any larvicidal efficacy effect of both types of aqueous extracts (*A. indica* leaves and *C. colocythis* fruit) was noticed very insignificant as compared to AgNPs mediated by these extracts. Some other researchers have also reported negligible biocidal activity of aqueous extracts in comparison to extracts mediated nanoparticles [21, 23, 47]. On the other hands, a concentration-dependent biocidal efficiency of pure aqueous extracts of various natural products can also be found in literature [16, 28]. It seems that biocidal activities of extracts depend not only on the concentrations used in the bioassays but also on the level of purification (crude, highly purified, aqueous, or alcoholic-) as well as nature and different parts of bio-materials to be extracted such as leave, fruit, stem and seed [16, 19, 46].

In the case of extract-mediated AgNPs, the enhanced larvicidal activity of AgNPs can be attributed to the synergistic effect of Ag-ions and deposited biomolecules. The difference in the larvicidal efficiency occurs due to the presence of different types of biomolecules on the surfaces of AgNPs though Ag-ions were same in both cases. In both types of extract mediated AgNPs, We observed, in both cases, that by increasing the concentration of AgNPs, percent mortality of larvae increased. Nevertheless, *C. colocythis* fruit extract mediated AgNPs were found more effective at lower concentrations and exhibited strong larvicidal efficacy as compared to *A. Indica* leaves extract mediated AgNPs as demonstrated graphically in figure 13. As mentioned above, different types of biomaterials (plants) and even various parts of same plant may affect the biocidal efficiency of mediated AgNPs. That's why perhaps AgNPs prepared by fruit extract of *C. colocythis* showed better antilarval activity even at lower concentrations in our case. A comparative larvicidal study of extracts and mediated AgNPs using various parts such as leaves, fruit, seed and stem of same plant can be conducted for more detailed investigations.

Since the exact mechanism of larvae mortality is unidentified, different modes of action can be proposed for larvicidal activity of AgNPs. Most probably the size and surface chemistry of AgNPs play a vital role to make them lethal against mosquito larvae. At the first stage, owing to small size AgNPs penetrate into the larval membrane causing the possible leakage of cellular materials. In the next step, AgNPs may interact with the different cell molecules to inhibit molting and other physiological processes. In their action, AgNPs may produce peroxide, inactivate the enzymes and disturb the functional proteins leading to the larvae death. The presence of extract biomolecules on the AgNPs surface can further enhance their effectiveness in this larvicidal process [23, 40, 46, 47]. This indicates that anti-dengue antilarval pharmaceuticals can be formulated using very low concentration of natural products based AgNPs for practical applications. Thus findings of this study can be helpful to provide new paradigm in designing the green chemistry based alternative to combat the increasing mosquitoes' diseases.

5. Conclusion

AgNPs were prepared from herbal origin which is economical and eco-friendly using the aqueous extracts of *A. indica* leaves and *C. colocythis* fruit. Both types of NPs were found pure metallic silver in nature having FCC crystalline structure and were spherical in shape with diameters of 25±5 nm, as determined by SEM and XRD. The optical behavior of colloidal samples and presence of different extract biomolecules were studied by UV- Vis, and FTIR. Both types of NPs were found pure metallic silver in nature having FCC crystalline structure and of spherical in shapes with diameters in the range of 25±5 nm, synthesized by different biomolecules of extracts as determined by different characterization techniques including SEM, UV- Vis, XRD and FTIR. Both types of The prepared green AgNPs exhibited different great larvicidal potential against *A. aegypti* even at low concentrations; LC₅₀ at 1.25 mg/Lppm for *A. indica* AgNPs and LC₅₀ at 0.3 mg/Lppm for *C. colocythis* mediated AgNPs respectively. We observed a concentration dependent larvicidal activity in both types of AgNPs whereas their extracts showed negligible efficiency against larvae in both cases. These results indicate suggest the potential use of green AgNPs as alternative to replace the synthetic products in pharmaceuticals for anti-dengue antilarval purposes applications.

Acknowledgments

Authors acknowledge Dr. Shahid Atiq and Dr. Syed Sajjad Hussain from Centre of Excellence in Solid State Physics, University of the Punjab, Lahore- Pakistan for their cooperation in experimental work and useful discussions in manuscript writing.

Ethical Statement

It is not relevant to our work.

Funding Statement

This work was financially supported by Higher Education Commission (HEC) of Pakistan under the Project of ‘National Research Program for Universities’ (Project No.: HEC-NRPU-8019).

Data Accessibility

Our data are deposited at Dryad: <https://datadryad.org/stash/dataset/doi:10.5061/dryad.z612jm68k> [48], with following reviewer URL; Dryad Reviewer URL: <https://datadryad.org/stash/share/DksW82FCDqISwPqI2aYhDI4C9QUE74hEBPTmrJ3trHW>

Competing Interests

We have no competing interests.

Authors' Contributions

- 1). SR and MAR performed the synthesis experiments, analysed the data and wrote the manuscript
- 2). ZK, SR and MAR designed the experiments and conducted the larvicidal bioassays
- 3). SR and SN conducted the characterization measurements and helped in interpretation of data
- 4). FM and ZK provided the larvae facility and helped in larvicidal activity experiments
- 5). SR and SN conducted the characterization measurements and helped in interpretation of data
- 6). MJR and SR conducted the FTIR and SEM characterization and helped in data analysis

References

[revised manuscript text omitted]

Figure 11. Graphs showing the larvicidal effect of AgNPs at different concentrations (a) *A. indica* AgNPs showed LC_{50} at about 1.25 mg/L while at 20 mg/L 100% mortality was observed, (b) *C.colocynthis* AgNPs exhibited LC_{50} at 0.3 mg/L and 100% mortality was observed at at 5 mg/L.

Figures

Figure 1

Figure 2

Figure 53

Figure 64

Figure 75

Figure 86

Figure 97

Figure 108

Figure 119

Figure 1210

Figure 1311

Reply to Reviewer's Reports

(Manuscript ID RSOS-200540)

We are thankful to the reviewers for their expert assessment of our manuscript. We are grateful for their thoughtful and valuable comments to improve our manuscript. All points and issues raised by the learnt reviewers have been considered and the manuscript has been modified accordingly. A detailed point-by-point response to all comments is provided below indicating the implemented changes in in the revised version of the manuscript. A highlighted version by 'Track Changes' is also included with the resubmission.

Response to 1st Reviewer's Comments

Comment 1. The article describes the synthesis of Ag nanoparticles using *Azadirachta indica* and *Citrullus colocynthis* extracts. The work done is useful and is in line with the current scientific demands. There are also some issues to be addressed before publishing this work.

Response 1. We are grateful to the learnt reviewere for mentioning our work "*useful*" and "*in line with current scientific demands*".

Comment 2. There are several grammatical mistakes for instance I am highlighting some of these:

- i) page 2 line 17 which have famous is incorrect it should be which are famous
- ii) page 2 line 25 "vector for many deceases like" it should be vector for many diseases like
- iii)page 2 line 25 please correct "fever.Dengue"
- iv) page 2 line 35 "AgNPs were manufactured" were synthesized
- v) page 2 line 35 "the mixture was let to cool down to room temperature" should be the mixture was cooled to room temperature

The authors should thoroughly check the whole manuscript and English should be improved as well as all the editorial mistakes should be rectified.

**Response 2.** We are thankful to the reviewer for pointing out these grammatical mistakes to
improve beauty of the text. The whole manuscript has been thoroughly checked
and all the grammatical and editorial mistakes mentioned by the reviewers and
otherwise are corrected.

**Comment 3.** Page 6 line 10, authors mentioned the particle size, I
guess it should be crystallite size?

**Response 3.** The corrections were made as suggested by the reviewer.

**Comment 4.** Authors have not mapped the particle size in SEM images.
The particle size should be measured in SEM image and it
should be visible in the picture.

**Response 4.** According to the reviewer recommendations, the SEM images have been
modified by adding the new images showing particle size visible in the picture.

**Comment 5.** The authors have mentioned that the chosen plants have
known biological activity. However, they do not show any
activity against *A. aegypti*. Secondly, if the extracts are
not active against *A. aegypti* why there is a large
difference in activity of AgNPs prepared from two
different extracts as LC50 at 1.25 ppm for *A. indica* and
LC50 at 0.3 ppm for *C. colocynthis* is observed. The later
is 4 times more active than the former one. I do not seem
that activity is only due to AgNPs in that case it should
be similar or close to each other.

**Response 5.** We are appreciative to the reviewer for highlighting this point.
Firstly, it is well-established fact that various medician plants including *A.*
*indica* and *C. colocynthis*, exhibit different biological activities (R1-R3).
However, in our case, both of the aqueous extracts (*A. indica* leaves and *C.*
*colocynthis* fruit) did not show any larvicidal efficacy against dengue vector.
The inefficiency of aqueous extracts of different plants was also reported by
other researchers [R3-R6] however, some studies report concentration-
dependent biological efficiency of aqueous extracts of some natural products
[R2,R7]. Thus, biological activity of extracts of biomaterials depend on various

factors such as nature, different parts (leave, fruit, stem and seed) and level of purification (crude, highly purified, aqueous, or alcoholic) and concentration [R8-R9].

The reason of the difference in larvicidal activity of AgNPs prepared by *A. indica* leaves or *C. colocynthis* fruit is that different types of biomolecules are deposited on the surface of AgNPs (as confirmed by FTIR analysis). This is, in fact, the synergistic effect of Ag-ions and biomolecules to enhance the larvicidal performance of extract mediated AgNPs. Thus the difference in larvicidal efficiency occurs due to the presence of different types of biomolecules on the surfaces of AgNPs although Ag-ions are same in both cases. Therefore, the LC₅₀ values for both cases were different (LC₅₀ at 1.25 ppm for *A. indica* and LC₅₀ at 0.3 ppm for *C. colocynthis*).

Main text has also been modified to explain this point.

Comment 6. Figures 1, 2 and 5 are unnecessary and should be deleted.

Response 6. Figures 1 and 3 and figures 2 and 4 are combined according to the recommendations of reviewer 2, instead of deleting. Figure 5 depicts schematics of the potential formation mechanism of extract mediated AgNPs and different phases of nucleation and growth processes. We would like to keep it in the main text with the kind consent of our proficient reviewer.

Data of all modified figures has been uploaded to Dryad Digital Repository

Our data are deposited at Dryad:

<https://datadryad.org/stash/dataset/doi:10.5061/dryad.z612jm68k> [

Dryad Reviewer URL:

<https://datadryad.org/stash/share/DksW82FCDqISwPqI2aYhDI4C9QUE74hEBPTmrJ3trHw>

Comment 7. the sentence in conclusion section “Both types of NPs were found pure metallic silver in nature having FCC crystalline structure and of spherical in shapes with diameters in the range of 25±5 nm, synthesized by different biomolecules of extracts as determined by different characterization techniques including SEM, UV-Vis, XRD and FTIR” is too long and understandable. It should be refabricated to two sentences. After addressing these issues the article can be published

in RSOS.

**Response 7.** We are agreed with the reviewer and this sentence has been rephrased into two
sentences as,

“Both types of NPs were found pure metallic silver in nature having FCC
crystalline structure and were spherical in shapes with diameters in the range of
25±5 nm, as determined by SEM and XRD. The optical behavior of colloidal
samples and presence of different extract biomolecules were studied by UV-
Vis, and FTIR.” The main text has also been modified accordingly

We are obliged to the reviewer for recommending our manuscript for “
*publication*” in RSOS.

Response to 2nd Reviewer’s Comments

**Comment 1.** The MS presents work which has been published with various
other plants. The work carried out is significant but need
major revision.

**Response 1.** We are obliged to the reviewer for mentioning our work “*significant*”.

**Comment 2.** MS needs enormous revision in scientific reporting,
language and grammar. The scientific names are wrongly
spelled at some places.

**Response 2.** We are grateful to the learnt reviewer for highlighting the errors to enhance the
flouncy and beauty of the text. The manuscript is carefully examined and all
mistakes of language, grammar, typos and scientific reporting including
scientific names pointed out by the reviewer and otherwise are corrected and
highlighted by ‘Track Changes’ in the revised manuscript.

**Comment 3.** Each section of the MS needs major revision. The comments
have been marked on the MS. Authors are advised to go
through each and every comment; and rectify accordingly.

**Response 3.** We are thankful to the reviewer for valuable comments and suggestions. We
have addressed all the comments and points raised by the learnt reviewer and

manuscript has been thoroughly revised in the light of remarks and
recommendations of the reviewer.

The detailed response to the comments is described below to highlight the
amendments made in the revised version of the manuscript accordingly.

**Comment:** The MS is about anti-dengue vector. Kindly revise.

**Response :** The title of the MS has been rectified as directed by the reviewer.

**Comment:** Check series.... FTIR was done in last..Please write based
on conducting the characterization

**Response :** In abstract, series of the characterizations techniques has been corrected, as
suggested by the reviewer.

**Comment:** Authors have not added any comparison between the
characteristics of two kinds of NPs. Some features may be
mentioned.

**Response :** We agreed to the reviewer and the abstract has been modified by adding
comparative characteristic features of both types of AgNPs.

**Comment:** write scientific names correctly throughout the
manuscript. Start specific names with small case. Take
care of spellings. *A. indica* and *C. colocynthis*

**Response:** All scientific names and terms have been corrected including *A. indica* and *C.*
*colocynthis*, as directed by the reviewer.

**Comment:** Which part? As you have prepared NPs with fruits, please
specify

**Response :** The *C. colocynthis* fruit has been specified and reference has also been added in
the main text, as suggested by the reviewer.

**Comment:** Dengue is transmitted by *Aedes* mosquitoes. It is caused by
virus. Larvae have no role in it. Please rectify Write as
hyphenated throughout the MS

**Response:** We agreed to the reviewer and the sentence has been rectified in the main text
as; “Dengue fever incidence has increased 30-fold worldwide in the last 30
5 years which is caused by dengue virus”

**Comment:** What do you mean by glass substrate?

**Response :** Glass substrate means just a small piece of glass slide on which few drops of
colloidal AgNPs sample are dried to make a thick enough NPs layer. This was
used as sample for characterization purposes. It has also been described in
literature [R10, R11].

**Comment:** 1ml, 2.5 ml and 4ml..... on what basis the ratios
(pattern of conc.) were decided? The amount/conc. of a
compound to be used in a mixture is always in a particular
ratio. For example...1, 2, 4, 8....

However, only AgNPs prepared by using the 500 μ l of
extract was used in larvicidal activity and other
characterization, specify the reason plz

**Response:** In this study, two types of extracts were used as reducing and stabilizing agents
for synthesis of AgNP. The amount of extracts in both cases was selected from
the literature protocols [R12, R13]. To obtain the optimized properties three
different amounts of each extract were used for the preparation of AgNPs. An
amount was selected and then its 2 fold and 4 fold was used for AgNPs
synthesis process. In the case of *A. indica* extract, 250 μ L, 500 μ L and 1000 μ L
while for *C. colocynthis* extract 1 mL, 2 mL and 4 mL were used (2.5 mL was
typo error which has been corrected in the main text). The AgNPs prepared
from 500 μ L (*A. indica*) and 2 mL (*C. colocynthis*) were taken as optimized
AgNPs on the basis of Uv-Vis results. This point has been addressed in the main
text according to the reviewer’s recommendations.

**Comment:** Why did authors use directly 5 mM concentration to prepare
NPs? Higher concentration of silver nitrate itself is
toxic? Did you try for lower concentrations? What was the
result? If not, then why?

**Response:** We agreed to the reviewer that AgNPs can be made by different concentrations
of precursor AgNO₃ solution (Lower and higher than 5mM). In this study, both
types of AgNPs were prepared following the protocols mentioned in the
literature. *A. indica* extract-mediated AgNPs were synthesized using 1mM
solution of AgNO₃ using methods described by of Aparajita et al. [R12] while
*C. colocynthis* extract-mediated AgNPs using 5mM concentration of precursor
solution as reported by Shawkey et al. [R13].

**Comment:** We cannot say that the fruit extract was more effective at
lower concentration as the comparison is between 1 mM and
5 mM of AgNO₃. 5 mM is itself a very high concentration of
silver nitrate. If authors want to compare the efficacy
then they should do at similar concentrations either 1 mM
or 5 mM

**Response:** In our work we used two different types of extracts and AgNPs to study the
larvicidal activity. Each type of AgNPs were prepared following different
protocols using different precursor concentration (1mM or 5mM) and two
different reducing agents (*A. indica leaves* extract or *C. colocynthis fruit*
extract). We aim to report the antilarvae performance of each type of extract and
AgNPs at different concentrations. It is well-known fact that change in
concentration of precursor solution will affect the size and morphology of
prepared AgNPs [R14, R15]. In our case, we used two type of extracts as
reducing and stabilizing agents and thus different types of biomolecules were
deposited on the surface of each type of AgNPs which caused variation in
larvicidal efficacy.

We have rectified the main text (abstract, result and discussion and conclusions)
to address the comparison issue raised by the reviewers.

**Comment:** ppm unit is no longer in use. May be replaced

**Response:** According to reviewer recommendations, ppm has been replaced by mg/L
throughout the manuscript.

**Comment:** "The test concentrations of both types of AgNPs were
selected on the basis of pilot experiments in which LC50

concentrations were found”.....Any published result? If
so, give ref. Otherwise, you may give values obtained here

**Response:** The LC₅₀ values of pilot experiments were found to be 1.25 mg/L and 0.3 mg/L
for *A. indica* and *C. colocynthis* mediated AgNPs respectively. These LC₅₀
values have also been mentioned in the main text according to the reviewer
suggestions .

**Comment:** This time dependent color change during the synthesis,
actually,time not mentioned

**Response:** Time has been mentioned in the main text as “10 min for *A. indica* mediated
AgNPs and 20 min for *C. colocynthis* mediated AgNPs.”

**Comment:** we can notice strong silver signal along with some other
signals. mention Ag%

**Response:** % age values of Ag have been mentioned for both types of AgNPs in main text
as “37.08 % for *A. indica* AgNPs and 38.5% for *C. colocynthis* AgNPs” as
directed by the reviewer.

**Comment:** mL denotes the volume. As per WHO protocol, 1 mL of
toxicant is added to 249 mL of water to study larval
mortality. What about concentration? Write the conc of
extract. How can you compare the mL of extract with ppm of
AgNPs?

**Response:** The reviewer is right that ml determines the volume only. We have added
percentage concentration of the extract. It is in common practice to use the
extract in percentage concentration [R16, R17]. The prepared extract is
standardized as 100 % from which dilutions are drawn for biassays. We used 3
different amounts (1mL, 2mL and 3mL) of the extract to be dissolved in water
to make a final volume of 200 mL. 1 mL of extract in 200 mL of water will
make a 0.5 % extract concentration, 2 mL will make 1 % and 3 mL will make
1.5 % extract concentration in the solution. Furthermore because the dynamics
of the concentration formulations for extract and nanoparticles is different,
extract is based on the extracted material while nanoparticles' concentration is

devised from the precursor used, therefore, it is not necessary to have a direct
comparison of the two in terms of concentration units.

**Comment:** 1ml, 2ml, 4 ml..... Are these concentrations? , 2, 3
10 mL... Is this concentration?

**Response:** No, these are quantities, now we have added % age concentration of extracts (
0.5 %, 1%, and 1.5%) in main text to rectify this issue raised by the reviewer.

**Comment:** How can you make anti-dengue medicine which has been
tested against dengue larvae not dengue virus

**Response:** We are thankful to the reviewer for indicating this point, we have rectified the
main main text as “This indicates that antilarval pharmaceuticals can be
formulated using very low concentration of natural products based AgNPs for
practical applications”

**Comment:** Figure captions: all figure captions need to be
revised...as they too lengthy, unnecessarily too big
legends. Please make all legends concise and crisp.

**Response:** All the figure captions have been revised to make brief and concise, as directed
by reviewer.

**Comment:** Figs. 1, 2 and 3 can be easily combined. Please shorten
and make it crisper. F1: explanations can be included in
the figures itself (labellings of figures)

**Response:** According to the reviewer recommendations, previous figure 1 and 3 have been
combined into one (new figure no. 1) and figure 2 has been combined with
figure 4 to make new figure no. 2. The main text has also been amended
accordingly.

**Comment:** F11 and F12: Please delete...instead add LC values tables

Response: The LC values have already been added in tables 1 and 2. The figures 11 and 12 provide the visual demonstration of the larvicidal activity, so with the kind consent of our proficient reviewer we would like to keep them in the main text.

References:

- R1. Abu-Darwish MS, Efferth T. 2018 Medicinal plants from near east for cancer therapy. *Front. Pharmacol.* 9, 56. (doi: 10.3389/fphar.2018.00056)
- R2. Kadir SLA, Yaakob H, Mohamed RZ. 2013 Potential anti-dengue medicinal plants: a review. *J Nat Med* 67:677–689 (doi: 10.1007/s11418-013-0767-y)
- R3. Govindachari TR, Suresh G, Gopalakrishnan G, Banumathy B, Masilamani S. 1998 Identification of antifungal compounds from the seed oil of *Azadirachta indica*. *Phytoparasitica* 26, 109–116. (doi: 10.1007/BF02980677)
- R4. Shawkey AM, Abdulall AK, Rabeh MA, Abdellatif AO. 2014 Enhanced biocidal activities of *Citrullus colocynthis* aqueous extracts by green nanotechnology. *International Journal of Applied Research in Natural Products* 7 (2), 1-10.
- R5. Shawkey AM, Rabeh MA, Abdulall AK, Abdellatif AO. 2013 Green nanotechnology: Anticancer activity of silver nanoparticles using *Citrullus colocynthis* aqueous extracts. *Adv. Life Sci. Technol.* 13, 60–70.
- R6. Rawani A, Ghosh A, Chandra G. 2013 Mosquito larvicidal and antimicrobial activity of synthesized nano-crystalline silver particles using leaves and green berry extract of *Solanum nigrum* L. (Solanaceae: Solanales). *Acta Tropica* 128, 613–622. (doi: 10.1016/j.actatropica.2013.09.00)
- R7. Sujitha V, Murugan K, Paulpandi M, Panneerselvam C, Suresh U, Roni M, Nicoletti M, Higuchi A, Madhiyazhagan P, Subramaniam J, Dinesh D, Vadivalagan C, Chandramohan B, Alarfaj AA, Munusamy MA, Barnard DR, Benelli G. 2015 Green-synthesized silver nanoparticles as a novel control tool against dengue virus (DEN-2) and its primary vector *Aedes aegypti*. *Parasitol Res.*, 114, 3315-25. (doi: 10.1007/s00436-015-4556-2)
- R8. Suganya G, Karthi S, Shivakumar MS. 2014 Larvicidal potential of silver nanoparticles synthesized from *Leucas aspera* leaf extracts against dengue vector *Aedes aegypti*. *Parasitol Res.* 113, 875–880. (doi: 10.1007/s00436-013-3718-3)
- R9. Suresh U, Murugan K, Benelli G, Nicoletti M, Barnard DR, Panneerselvam C, Kumar PM, Subramaniam J, Dinesh D, Chandramohan B. 2015 Tackling the growing threat of dengue: *Phyllanthus niruri*-mediated synthesis of silver nanoparticles and their mosquitocidal properties against the dengue vector *Aedes aegypti* (Diptera: Culicidae). *Parasitol Res.* 114(4), 1551-1562. (doi: 10.1007/s00436-015-4339-9)

R10. Kanwal Z, Raza MA, Riaz S, Manzoor S, Tayyeb A, Sajid I, Naseem S. 2019
Synthesis and characterization of silver nanoparticle-decorated cobalt
nanocomposites (Co@AgNPs) and their density-dependent antibacterial activity. *R.*
*Soc. open sci.* **6**, 182135. (doi:10.1098/rsos.182135)
R11. Kanwal Z, Raza, MA, Manzoor F, Riaz S, Jabeen G, Fatima S, Naseem S. 2019 A
Comparative Assessment of Nanotoxicity Induced by Metal (Silver, Nickel) and
Metal Oxide (Cobalt, Chromium) Nanoparticles in *Labeo rohita*. *Nanomaterials*, **9**,
309. (doi:10.3390/nano9020309)
R12. 22-Aparajita V, Mohan SM. 2016 Controllable synthesis of silver nanoparticles
using Neem leaves and their antimicrobial activity. *J Radiat Res Appl Sc* **9(1)**, 109-
115. (doi: 10.1016/j.jrras.2015.11.001)
R13. 23-Shawkey AM, Rabeh MA, Abdulall AK, Abdellatif AO. 2013 Green
nanotechnology: Anticancer activity of silver nanoparticles using *Citrullus*
*colocynthis* aqueous extracts. *Adv. Life Sci. Technol.* **13**, 60–70.
R14. Devadiga A, Vidya SK, Saidutta MB. 2017 Effect of Precursor Salt Solution
Concentration on the Size of Silver Nanoparticles Synthesized Using Aqueous Leaf
Extracts of *T. catappa* and *T. grandis* Linn f.—A Green Synthesis Route. In: Mohan B.
R., Srinikethan G., Meikap B. (eds) *Materials, Energy and Environment*
*Engineering*. Springer, Singapore. (doi: 10.1007/978-981-10-2675-1_17)
R15. Vishwakarma K, Shweta, Upadhyay N, Singh J, Liu S, Singh VP, Prasad SM,
Chauhan DK, Tripathi DK, Sharma S. 2017 Differential Phytotoxic Impact of Plant
Mediated Silver Nanoparticles (AgNPs) and Silver Nitrate (AgNO₃) on *Brassica sp.*
*Front Plant Sci* **8**, 1501. (doi: 10.3389/fpls.2017.01501)
R16. Subramaniam K, Siswomihardjo W, Sunarintyas S. 2005 The effect of different
concentrations of Neem (*Azadiractha indica*) leaves extract on the inhibition of
*Streptococcus mutans* (In vitro). *Maj Ked Gigi (Dent J)* **38**, 176-9. ..
R17. Ederley V, Gloria C, Gladis M, César H, Jaime O, Oscar A. 2018 Silver
Nanoparticles Obtained by Aqueous or Ethanolic *Aloe vera* Extracts: An Assessment
of the Antibacterial Activity and Mercury Removal Capability. *J Nanomater* 2018,
Article ID 7215210, 7 pages (doi: 10.1155/2018/7215210)

Upper panel-Different steps during the preparation of aqueous extract of *A. indica* leaves; Middle Panel- Synthesis of AgNPs; Lower Panel-; Lower Panel- (h) precursor solution, (i) reducing agent, (j-l) colloidal samples of AgNPs prepared by different amounts of *A.indica* extract.

229x252mm (150 x 150 DPI)

Upper panel-Different steps during the preparation of aqueous extract of *C. colocynthis* leaves; Middle Panel- Synthesis of AgNPs; Lower Panel- (h) precursor solution, (i) reducing agent, (j-l) colloidal samples of AgNPs prepared by different amounts of *C. colocynthis* extract.

257x248mm (150 x 150 DPI)

Schematic illustration of the synthesis mechanism of extract-mediated AgNPs. The phenomena occurring in each phase is mentioned accordingly.

286x153mm (150 x 150 DPI)

UV-Vis spectroscopy results of AgNPs synthesized by (a) three different concentrations of *A. indica* leaves extract and (b) three different concentrations of *C. colocynthis* fruit extract. Single absorption peak in each case appeared in the wavelength range of 412-430 nm indicating the spherical shaped AgNPs. Insets display images of each colloidal sample in cuvette.

474x188mm (150 x 150 DPI)

SEM micrographs and histograms. (a,b) SEM image of *A. indica* AgNPs showed spherical shaped NPs decorated on larger extract particles, (c) Histogram of *A. indica* AgNPs indicated their diameter ranging between 12-24 nm. (d,e) SEM results of *C. colocynthis* AgNPs reveal the sphere like morphology and random dispersal over the bigger extract particles. (f) Histogram of *C. colocynthis* AgNPs indicated their diameter ranging between 20-36 nm

321x187mm (150 x 150 DPI)

EDX spectrum showing elemental composition analysis of AgNPs (a) *A. indica* leaves, (b) *C. colocynthis* fruit. Insets show the table of elements in each case.

256x303mm (150 x 150 DPI)

X-ray diffraction patterns of (a) *A. indica* AgNPs and (b) *C. colocynthis* AgNPs

476x190mm (150 x 150 DPI)

FTIR spectra (a) *A. indica* AgNPs, (b) *C. colocynthis* AgNPs. Occurrence of various absorption peaks at different positions indicates the presence of different biomolecules on the surface of NPs.

376x152mm (150 x 150 DPI)

A representative display of larvicidal bioassay. Water (control), *A. indica* extract (Conc: 1.5 %), AgNPs at
different concentrations. Upper panel shows the initial stage of experiment with 0 h time. Lower panel shows
results after 24 h.

370x118mm (150 x 150 DPI)

A typical demonstration of larvicidal activity. Water (control), *C. colocynthis* extract (Conc: 1.5 %), AgNPs at
different concentrations. Upper panel shows stage in the start of the experiment (at 0 h). Lower panel
displays the final stage story (after 24 h).

370x118mm (150 x 150 DPI)

Graphs showing the larvicidal effect of AgNPs at different concentrations (a) *A. indica* AgNPs showed LC50 at about 1.25 mg/L while at 20 mg/L 100% mortality was observed, (b) *C. colocynthis* AgNPs exhibited LC50 at 0.3 mg/L and 100% mortality was observed at 5 mg/L.

283x74mm (150 x 150 DPI)

Appendix D

Reply to Reviewer's Reports (Manuscript ID RSOS-200540)

We are again grateful to the reviewers for their skillful suggestions and valuable comments to improve our manuscript. All minor points and issues raised by the learnt reviewers have completely been addressed and the manuscript has been modified accordingly. A detailed point-by-point response to all comments is also provided below indicating the implemented changes in the revised version of the manuscript. A highlighted version by 'Track Changes' is also included with the resubmission.

Response to Comments Reviewer # 2

Comment: I appreciate the efforts taken in revision of the manuscript. A few errors are yet to be addressed. The errors have been incorporated in the MS. Authors are requested to go through the MS and address them.

Response: We are obliged to the expert reviewer for appreciating our efforts in revised manuscript. All the issues and errors highlighted by the reviewer are addressed and the main text has been modified accordingly.

Comment: "only AgNPs prepared by using the 500 μ L of extract were taken as optimized NPs" Can you provide a reason for choosing this extract concentration

Response: We chose 50 mL *A. indica* extract mediated AgNPs for bioassay on the basis of UV-Vis spectroscopy results. It can be seen from Uv-Vis spectra (Figure 4a) that the broadening of absorption peaks for 250 μ L and 1000 μ L extract mediated AgNPs were larger as compared to 500 μ L AgNPs. Narrow peak width qualitatively indicates monodispersity and uniformity in the size distribution of NPs. This has also been mentioned in the main text

Comment: Can you provide a reason for choosing this particular (2 mL extract mediated) NPs

Response : In the case of *C.colocynthis* extract mediated AgNPs, again on the basis of Uv-Vis spectra, 2 mL extract mediated AgNPs were selected for larvicidal activity. Because of strong absorption intensity and narrow peak width as compared to 1 mL and 4 mL extracted mediated AgNPs respectively. The main text has also been modified to rectify this point.